# Simultaneous Localization and Mapping (SLAM) for Autonomous Driving: Concept and Analysis

**Shuran Zheng [1],\*, Jinling Wang [1] , Chris Rizos [1] , Weidong Ding [1] and Ahmed El-Mowafy [2]**

1 School of Civil and Environmental Engineering, UNSW Sydney, Sydney 2052, Australia
2 School of Earth and Planetary Sciences, Curtin University, Perth 6845, Australia
\* Correspondence: shuran.zheng@unsw.edu.au

**Abstract:** The Simultaneous Localization and Mapping (SLAM) technique has achieved astonishing progress over the last few decades and has generated considerable interest in the autonomous driving community. With its conceptual roots in navigation and mapping, SLAM outperforms some traditional positioning and localization techniques since it can support more reliable and robust localization, planning, and controlling to meet some key criteria for autonomous driving. In this study the authors first give an overview of the different SLAM implementation approaches and then discuss the applications of SLAM for autonomous driving with respect to different driving scenarios, vehicle system components and the characteristics of the SLAM approaches. The authors then discuss some challenging issues and current solutions when applying SLAM for autonomous driving. Some quantitative quality analysis means to evaluate the characteristics and performance of SLAM systems and to monitor the risk in SLAM estimation are reviewed. In addition, this study describes a real-world road test to demonstrate a multi-sensor-based modernized SLAM procedure for autonomous driving. The numerical results show that a high-precision 3D point cloud map can be generated by the SLAM procedure with the integration of Lidar and GNSS/INS. Online four–five cm accuracy localization solution can be achieved based on this pre-generated map and online Lidar scan matching with a tightly fused inertial system.

**Keywords:** Simultaneous Localization and Mapping; autonomous driving; localization; high definition map

## 1. Introduction

Autonomous (also called self-driving, driverless, or robotic) vehicle operation is a significant academic as well as an industrial research topic. It is predicted that fully autonomous vehicles will become an important part of total vehicle sales in the next decades. The promotion of autonomous vehicles draws attention to the many advantages, such as service for disabled or elderly persons, reduction in driver stress and costs, reduction in road accidents, elimination of the need for conventional public transit services, etc. [1,2].

A typical autonomous vehicle system contains four key parts: localization, perception, planning, and controlling (Figure 1). Positioning is the process of obtaining a (moving or static) object's coordinates with respect to a given coordinate system. The coordinate system may be a local coordinate system or a geodetic datum such as WGS84. Localization is a process of estimating the carrier's pose (position and attitude) in relation to a reference frame or a map. The perception system monitors the road environment around the host vehicle and identifies interested objects such as pedestrians, other vehicles, traffic lights, signage, etc.

By determining the coordinates of objects in the surrounding environment a map can be generated. This process is known as Mapping.

Path planning is the step that utilizes localization, mapping, and perception information to determine the optimal path in subsequent driving epochs, guiding the automated

vehicle from one location to another location. This plan is then converted into action using the controlling system components, e.g., brake control before the detected traffic lights, etc.

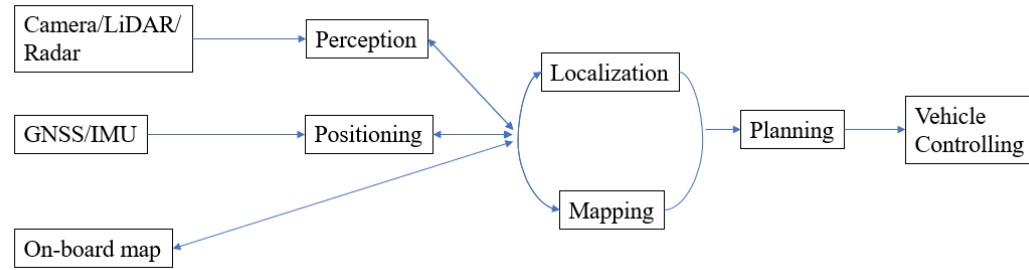

**Figure 1.** Functional components of an autonomous driving system.

All these parts are closely related. The location information for both vehicle and road entities can be obtained by combining the position, perception, and map information. In contrast, localization and mapping can be used to support better perception. Accurate localization and perception information is essential for correct planning and controlling.

To achieve fully automated driving, there are some key requirements that need to be considered for the localization and perception steps. The first is accuracy. For autonomous driving, the information about where the road is and where the vehicle is within the lane supports the planning and controlling steps. To realize these, and to ensure vehicle safety, there is a stringent requirement for position estimation at the lane level, or even the "where-in-lane" level (i.e., the sub-lane level). Recognition range is important because the planning and controlling steps need enough processing time for the vehicle to react [3]. Robustness means the localization and perception should be robust to any changes while driving, such as driving scenarios (urban, highway, tunnel, rural, etc.), lighting conditions, weather, etc.

Traditional vehicle localization and perception techniques cannot meet all of the aforementioned requirements. For instance, GNSS error occurs as the signals may be distorted, or even blocked, by trees, urban canyons, tunnels, etc. Often an inertial navigation system (INS) is used to support navigation during GNSS signal outages, to continue providing position, velocity, and altitude information. However, inertial measurement bias needs frequently estimated corrections or calibration, which is best achieved using GNSS measurements. Nevertheless, an integrated GNSS/INS system is still not sufficient since highly automated driving requires not only positioning information of the host vehicle, but also the spatial characteristics of the objects in the surrounding environment. Hence perceptive sensors, such as Lidar and Cameras, are often used for both localization and perception. Lidar can acquire a 3D point cloud directly and map the environment, with the aid of GNSS and INS, to an accuracy that can reach the centimeter level in urban road driving conditions [4]. However, the high cost has limited the commercial adoption of Lidar systems in vehicles. Furthermore, its accuracy is influenced by weather (such as rain) and lighting conditions. Compared to Lidar, Camera systems have lower accuracy but are also affected by numerous error sources [5,6]. Nevertheless, they are much cheaper, smaller in size, require less maintenance, and use less energy. Vision-based systems can provide abundant environment information, similar to what human eyes can perceive, and the data can be fused with other sensors to determine the location of detected features.

A map with rich road environment information is essential for the aforementioned sensors to achieve accurate and robust localization and perception. Pre-stored road information makes autonomous driving robust to the changing environment and road dynamics. The recognition range requirement can be satisfied since an onboard map can provide timely information on the road network. Map-based localization and navigation have been studied using different types of map information. Google Map is one example as it provides worldwide map information including images, topographic details, and satellite images [7], and it is available via mobile phone and vehicle apps. However, the use of maps will be limited by the accuracy of the maps, and in some selected areas the map's resolution may be inadequate. In [8], the authors considered low-accuracy maps for navigation by combining

data from other sensors. They detected moving objects using Lidar data and utilized a GNSS/INS system with a coarse open-source GIS map. Their results show their fusion technique can successfully detect and track moving objects. A precise curb-map-based localization method that uses a 3D-Lidar sensor and a high-precision map is proposed in [9]. However, this method will fail when curb information is lacking, or obstructed.

Recently, so-called "high-definition" (HD) maps have received considerable interest in the context of autonomous driving since they contain very accurate, and large volumes of, road network information [10]. According to some major players in the commercial HD map market, 10–20 cm accuracy has been achieved [11,12], and it is predicted that in the next generation of HD maps, a few centimeters of accuracy will be reached. Such maps contain considerable information on road features, not only the static road entities and road geometry (curvature, grades, etc.), but also traffic management information such as traffic signs, traffic lights, speed limits, road markings, and so on. The autonomous car can use the HD map to precisely locate the host-car within the road lane and to estimate the relative location of the car with respect to road objects by matching the landmarks which are recognized by onboard sensors with pre-stored information within the HD map.

Therefore, maps, especially HD maps, play several roles in support of autonomous driving and may be able to meet the stringent requirements of accuracy, precision, recognition ranging, robustness, and information richness. However, the application of the "map" for autonomous driving is also facilitated by techniques such as Simultaneous Localization and Mapping (SLAM). SLAM is a process by which a moving platform builds a map of the environment and uses that map to deduce its location at the same time. SLAM, which is widely used in the robotic field, has been demonstrated [13,14] as being applicable for autonomous vehicle operations as it can support not only accurate map generation but also online localization within a previously generated map.

With appropriate sensor information (perception data, absolute and dead reckoning position information), a high-density and accurate map can be generated offline by SLAM. When driving, the self-driving car can locate itself within the pre-stored map by matching the sensor data to the map. SLAM can also be used to address the problem of DATMO (detection and tracking of moving objects) [15] which is important for detecting pedestrians or other moving objects. As the static parts of the environment are localized and mapped by SLAM, the dynamic components can concurrently be detected and tracked relative to the static objects or features. However, SLAM also has some challenging issues when applied to autonomous driving applications. For instance, "loop closure" can be used to reduce the accumulated bias within SLAM estimation in indoor or urban scenarios, but it is not normally applicable to highway scenarios.

This paper will review some key techniques for SLAM, the application of SLAM for autonomous driving, and suitable SLAM techniques related to the applications. Section 2 gives a brief introduction to the principles and characteristics of some key SLAM techniques. Section 3 describes some potential applications of SLAM for autonomous driving. Some challenging issues in applying the SLAM technique for autonomous driving are discussed in Section 4. A real-world road test to show the performance of a multi-sensor-based SLAM procedure for autonomous driving is described in Section 5. The conclusions are given in Section 6.

## 2. Key SLAM Techniques

Since its initial introduction in 1986 [16], a variety of SLAM techniques have been developed. SLAM has its conceptual roots in geodesy and geospatial mapping [17].

In general, there are two types of approaches to SLAM estimation: filter-based and optimization-based. Both approaches estimate the vehicle pose states and map states at the same time. The vehicle pose includes 3D or 2D vehicle position, but sometimes also velocity, orientation or attitude, depending on the sensor(s) used and on the application(s).

### 2.1. Online and Offline SLAM

Figures 2 and 3 illustrate two general SLAM implementations: online SLAM and offline SLAM (sometimes referred to as full SLAM). According to [18], full SLAM seeks to calculate variables over the entire path along with the map, instead of just the current pose, while the online SLAM problem is solved by removing past poses from the full SLAM problem.

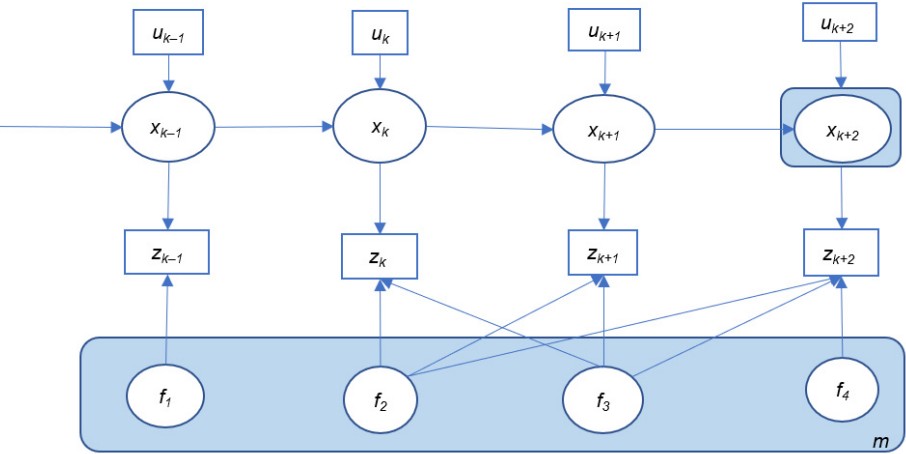

**Figure 2.** Description of online SLAM.

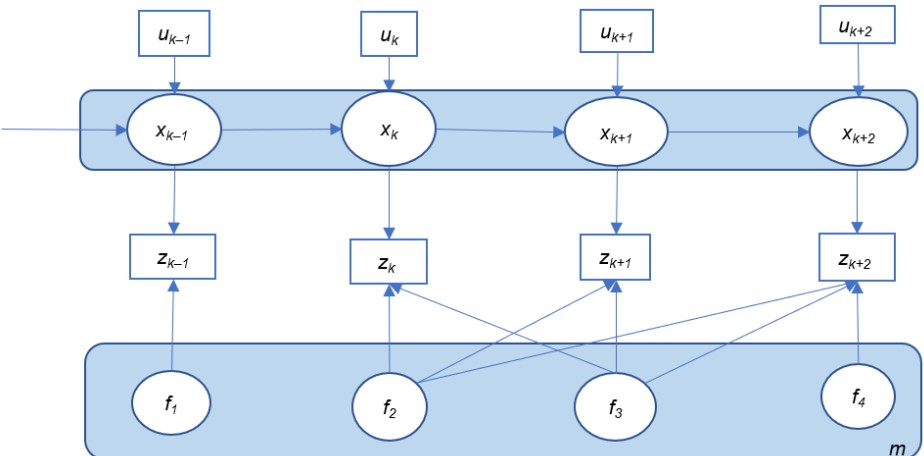

**Figure 3.** Description of offline SLAM.

Here, $x_k$ represents the vehicle pose (position, attitude, velocity, etc.) at time $k$. $m$ is the map that consists of stored landmarks ($f_1$–$f_4$) with their position states. $u_k$ is the control inputs that represent the vehicle motion information between time epochs $k - 1$ and $k$, such as acceleration, turn angle, etc., which can be acquired from vehicle motion sensors such as wheel encoders or an inertial sensor. At some epoch $k$, the onboard sensors (such as Camera, Lidar, and Radar) will perceive the environment and detect one or more landmarks. The relative observations between the vehicle and all the observed landmarks are denoted as $z_k$. With this information, the variables (including the vehicle pose and the map states) can be estimated.

The rectangle with blue background in Figures 2 and 3 represents the state variables that are estimated in these two implementations. In most cases, for online SLAM, only the current vehicle pose $x_{k+2}$ is estimated while the map is generated and updated with the most recent measurements ($u_{k+2}$ and $z_{k+2}$), whereas in the case of the offline SLAM implementation, the whole trajectory of the vehicle is updated together with the whole map. All the available control and observation measurements will be utilized together for the offline SLAM implementation.

However, with the development of SLAM algorithms and increased computational capabilities, the full SLAM solution may be obtained in real-time with an efficient SLAM algorithm, which can also be treated as an online problem. Therefore, implementing a SLAM method online or offline may be dependent on whether the measurement inputs (control and observation) it requires are from current/history or from future epochs, and on its processing time (real-time or not).

### 2.2. Filter-Based SLAM

Filter-based SLAM recursively solves the SLAM problem in two steps. Firstly, the vehicle and map states are predicted with processing models and control inputs. In the next step, a correction of the predicted state is carried out using the current sensor observations. Therefore, the filter-based SLAM is suitable for online SLAM.

Extended Kalman Filter-based SLAM (EKF-SLAM) represents a standard solution for the SLAM problem. It is derived from Bayesian filtering in which all variables are treated as Gaussian random variables. It consists of two steps: time update (prediction) and measurement update (filtering). At each time epoch the measurement and motion models are linearized (using the current state with the first-order Taylor expansion). However, since the linearization is not made around the true value of the state vector, but around the estimated value [19], the linearization error will accumulate and could cause a divergence of the estimation. Therefore, inconsistencies can occur.

Another issue related to EKF-SLAM is the continuous expansion of map size which makes the quadratic calculation process of large-scale SLAM impractical. For autonomous driving, the complex road environment and long driving period will introduce a large number of features, which makes real-time computation not feasible. A large number of algorithms have been developed in order to improve computational efficiency. For example, the Compressed Extended Kalman Filter (CEKF) [20] algorithm can significantly reduce computations by focusing on local areas and then extending the filtered information to the global map. Algorithms with sub-maps have also been used to address the computation issues [21–24]. A new blank map is used to replace the old map when the old one reaches a predefined map size. A higher-level map is maintained to track the link between each sub-map.

There are some other filter-based SLAM approaches, such as some variants of the Kalman Filter. One of them, the Information Filter (IF), is propagated with the inverse form of the state error covariance matrix, which makes this method more stable [25]. This method is more popular in multi-vehicle SLAM than in single-vehicle systems.

Another class of filter-based SLAM techniques is the Particle Filter (PF) which has become popular in recent years. PF executes Sequential Monte-Carlo (SMC) estimation by a set of random point clusters (or particles) representing the Bayesian aposteriori. The Rao–Blackwellized Particle Filter was proposed in [26]. Fast-SLAM is a popular implementation that treats the robot position distribution as a set of Rao–Blackwellized particles, and uses an EKF to maintain local maps. In this way, the computational complexity of SLAM is greatly reduced. Real-time application is possible with Fast-SLAM [27], making online SLAM possible for autonomous driving. Another advantage over EKF is that the particle filters can cope with non-linear motion models [28]. However, according to [29,30], Fast-SLAM suffers from degeneration since it cannot forget the past. If marginalizing the map and when resampling is performed, statistical accuracy is lost.

### 2.3. Optimization-Based SLAM

Full SLAM estimates all the vehicle pose and map states using the entire sensor data, and it is mostly optimization based. Similar to filter-based SLAM, the optimization-based SLAM system consists of two main parts: the frontend and the backend. In the frontend step, the SLAM system extracts the constraints of the problem with the sensor data, for example, by performing feature detection and matching, motion estimation, loop closure detection, etc. Nonlinear optimization is then applied to acquire the maximum likelihood estimation at the backend.

Graph SLAM is one of the main classes of full SLAM which uses a graphical structure to represent the Bayesian SLAM. All the platform poses along the whole trajectory and all the detected features are treated as nodes. Spatial constraints between poses are encoded in the edges between the nodes. These constraints result from observations, odometry measurements, and from loop closure constraints. After the graph construction, graph optimization is applied in order to optimize the graph model of the whole trajectory and map. To solve the full optimization and to calculate the Gaussian approximation of the aposteriori, a number of methods can be used, such as Gauss–Newton or Levenberg–Marquardt [31].

For graph-based SLAM, the size of its covariance matrix and update time are constant after generating the graph, therefore graph SLAM has become popular for building large-scale maps. Reducing the computational complexity of the optimization step has become one of the main research topics for practical implementations of the high-dimensional SLAM problem. The key to solving the optimization step efficiently is the sparsity of the normal matrix. The fact that each measurement is only associated with a very limited number of variables makes the matrix very sparse. With Cholesky factorization and QR factorization methods, the information matrix and measurement Jacobian matrix can be factorized efficiently, and hence the computational cost can be significantly reduced. Several algorithms have been proposed, such as TORO and g2o. The sub-map method is also a popular strategy for solving large-scale problems [32–36]. The sub-maps can be optimized independently and are related to a local coordinate frame. The sub-map coordinates can be treated as pose nodes, linked with motion constraints or loop closure constraints. Thus, a global pose graph is generated. In this way the computational complexity and update time will be improved.

Smoothing and Mapping (SAM), another optimization-based SLAM algorithm, is a type of nonlinear least squares problem. Such a least squares problem can be solved incrementally by Incremental Smoothing and Mapping (iSAM) [37] and iSAM2 [38]. Online SLAM can be obtained with incremental SAMs as they avoid unnecessary calculations with the entire covariance matrix. iSAM2 is more efficient as it uses a Bayes tree to obtain incremental variable re-ordering and fluid re-linearization.

SLAM++ is another incremental solution for nonlinear least squares optimization-based SLAM which is very efficient. Moreover, for online SLAM implementations, fast state covariance recovery is very important for data association, obtaining reduced state representations, active decision-making, and next best-view [39,40]. SLAM++ has an advantage as it allows for incremental covariance calculation which is faster than other implementations [40].

Table 1 is a summary of the characteristics of some typical SLAM techniques. Note that Graph SLAM utilizes all available observations and control information and can achieve very accurate and robust estimation results. It is suitable for offline applications and its performance relies on a good initial state guess. Filter-based SLAM is more suitable for small-scale environments when used for online estimation, but for the complex environment, a real-time computation may be difficult with the traditional EKF-SLAM. Other variants or fastSLAM should be considered. The incremental optimization method can do incremental updating, so as to provide an optimal estimation of a large-scale map with very high efficiency and in real-time.

**Table 1.** Characteristics of some typical SLAM techniques.

| SLAM | Type | Advantages | Disadvantages | Typical Studies |
|---|---|---|---|---|
| EKF SLAM | Bayesian filter | • Mature method, widely studied;<br>• Uncertainty is estimated. | • Suffers from linearization errors;<br>• No re-linearization step;<br>• Needs huge memory and computational resources for large maps. | [29,41,42] |

**Table 1.** *Cont.*

| SLAM | Type | Advantages | Disadvantages | Typical Studies |
|---|---|---|---|---|
| IF SLAM | Bayesian filter | • Already inversed covariance matrix; <br> • Faster and more stable than EKF; <br> • Suitable for multi-vehicle systems. | • Suffers from linearization errors; <br> • No re-linearization step. | [25,43] |
| CEKF SLAM | Bayesian filter | • Cost-effective; <br> • Outliers/errors only affect local maps; <br> • Auxiliary coefficient matrix is used for inactive parts. | • Needs correct link between local and global maps. | [20,44] |
| Fast SLAM | Particle filter | • Capable of updating with unknown data association; <br> • Less computation and memory cost than EKF; <br> • Suitable for nonlinear cases; <br> • Robust in cases where motion noise is high relative to measurement noise. | • Loses accuracy when marginalizing the map and resampling is performed. | [26,28,30,45] |
| Graph SLAM | Batch Least Squares optimization | • Suitable for nonlinear cases; <br> • More accurate; <br> • Can handle a large number of features. | • Not suitable for online applications; <br> • Relies on good initial value. | [46–51] |
| iSAM2 | Incremental optimization | • Very fast; <br> • Suitable for nonlinear cases; <br> • Allows re-linearization and data association correction. | • Complexity grows when graph become dense. | [37,38] |
| SLAM++ | Incremental optimization | • Suitable for nonlinear cases; <br> • Very fast estimation (faster than iSAM2); <br> • Efficient uncertainty estimation; <br> • Suitable for large-scale mapping. | • Complexity grows with increasing number of observations. | [39,40] |

*2.4. Sensors and Fusion Method for SLAM*

New SLAM methods have appeared thanks to advances in sensor and computing technology. These methods are also optimization-based or filtered-based at the backend estimation step while the frontend step is highly dependent on the application of different sensor modalities. Two of the major sensors used for SLAM are Lidar and Camera. The Lidar method has become popular due to its simplicity and accuracy compared to other sensors [52]. The core of Lidar-based localization and mapping is scan-matching, which recovers the relative position and orientation of two scans or point clouds. Popular approaches for scan matching include the Iterative Closet Point (ICP) algorithm and its variants [53–55], and the normal distribution transform (NDT) [56]. These methods are highly dependent on good initial guess, and are impacted by local minimums [57,58]. Some other matching methods include probabilistic methods such as correlative scan matching (CSM) [59], feature-based methods [57,60], and others. Many of the scan-matching methods focus on initial free of or robust to, initialization error, but they still face the computation efficiency challenge.

Some range sensors that can be used for SLAM estimation are Radar and Sonar/ultrasonic sensors. Radar works in a similar manner to Lidar, but the system emits radio waves instead of light to measure the distance to objects. Furthermore, since Radar can observe the relative velocity between the sensor and the object using the measured Doppler shift [61],

it is suitable for distinguishing between stationary and moving objects, and can be used to discard moving objects during the map-building process [62]. Some research on using Radar for SLAM can be found in [42,62–66]. When compared to Lidar, lower price, lower power consumption, and less sensitivity to atmospheric conditions make it well suited for outdoor applications. However, Radar has lower measurement resolution, and its detections are more sparse than Lidar. Thus, it is harder to match Radar data and deal with the data association problem, which results in its 3D mapping being less accurate.

Sonar/ultrasonic sensors also measure the time-of-flight (TOF) to determine the distance to objects, by sending and receiving sound waves. Sonar-based SLAM was initially used for underwater [67,68], and indoor [69] applications. It has become popular due to its low cost and low power consumption. It is not affected by visibility restrictions and can be used with multiple surface types [70]. However, similar to Radar, it obtains sparse information and suffers from inaccurate feature extraction and long processing time. Thus, it is of limited use for high-speed vehicle applications. Moreover, Sonar/ ultrasonic sensors have limited sensing range and may be affected by environmental noise and other platforms using ultrasound with the same frequency [71].

Camera is another popular sensor for SLAM. Different techniques have been developed, such as monocular [72,73], stereo [74–77], and multi-camera [78–81]. These techniques can be used in a wide range of environments, both indoor and outdoor. The single-camera system is easy to deploy, however, it suffers from scale uncertainty [82]. Stereo-camera systems can overcome the scale factor problem and can retrieve 3D structural information by comparing the same scene from two different perspectives [61]. Multi-camera systems have gained increasing interest, particularly as they achieve a large field of view [78] or are even capable of panoramic vision [81]. This system is more robust in complicated environments, while single sensor system may be very vulnerable to environmental interference [81]. However, the integration of Cameras requires additional software and hardware, and requires more calibration and synchronization effort [71,83]. Another special Camera, the RGB-D Camera, has been studied by the SLAM and computer vision communities [84–91] since it can directly obtain depth information. However, this system is mainly applicable in indoor environments because it uses infrared spectrum light and is therefore sensitive to external illumination [70].

The Visual SLAM can also be classified as feature-based or direct SLAM depending on how the measurements are used. The feature-based SLAM repeatedly detects features in images and utilizes descriptive features for tracking and depth estimation [92]. Some fundamental frameworks for this feature-based system include MonoSLAM [72,93], PTAM [94], ORB-SLAM [95], and ORB-SLAM2 [96]. Instead of using any feature detectors and descriptors, the direct SLAM method uses the whole image. Examples of direct SLAM include DTAM [97], LSD-SLAM [73], and SVO [98]. A dense or semi-dense environment model can be acquired by these methods, which makes them more computationally demanding than feature-based methods. Engel et al. [74] extended the LSD-SLAM from a monocular to a stereo model while Caruso et al. [99] extended the LSD-SLAM to an omnidirectional model. A detailed review of Visual SLAM can be found in [5] and [70,92,100,101].

Each of these perceptive sensors has its advantages and limitations. Lidar approaches can provide precise and long-range observations, but with limitations such as being sensitive to atmospheric conditions, being expensive, and currently rather bulky. Radar systems are relatively low cost, but are more suitable for object detection than for 3D map building. Sonar/ultrasonic sensors are not suitable for high-speed platform applications. Cameras are low-cost, even when multiple Cameras are used. Cameras can also provide rich visual information. However, they are sensitive to environment texture and light, and in general, have high computational demands. Therefore, a popular strategy is to combine a variety of sensors, making the SLAM system more robust.

There are several strategies to integrate data from different sensors for SLAM. One is fusing independently processed sensor results to then obtain the final solution. In [102], a mapping method that merged two grid maps, which were generated individually from

laser and stereo camera measurements, into a single grid map was proposed. In this method, the measurements of the different sensors need to be mapped to a joint reference system. In [103], a multi-sensor SLAM system that combined the 3-DoF pose estimation from laser readings, the 6-DoF pose estimation from a monocular visual system, and the inertial-based navigation estimation results to generate the final 6-DoF position using an EKF processing scheme was proposed. For this type of strategy, the sensors can provide redundancy and the system will be robust to possible single-sensor failure. A decision-making step may be needed to identify whether the data from each sensor is reliable, and to decide whether to adopt the estimation from that sensor modality or to ignore it. Another fusion strategy is using an assistant sensor to improve the performance of other sensor-based SLAM algorithms. The main sensor could be Lidar or Camera, while the assistant sensor could be any other type of sensor. In this strategy, the assistant sensor is used to overcome the limitations of the main sensor. The work in [104] incorporated visual information to provide a good initial guess on the rigid body transformation, and then used this initial transformation to seed the ICP framework. Huang et al. [105] extracted the depth of point-based and line-based landmarks from the Lidar data. The proposed system used this depth information to guide camera tracking and also to support the subsequent point-line bundle adjustment to further improve the estimation accuracy.

The above two strategies can be combined. In the work of [106], the fusing consists of two models, one deals with feature fusion that utilizes line feature information from an image to remove any "pseudo-segments", which result from dynamic objects, in the laser segments. Another is a modified EKF SLAM framework that incorporates the state estimates obtained from the individual monocular and laser SLAM in order to reduce the pose estimation covariance and improve localization accuracy. This modified SLAM framework can run even when one sensor fails since the sensor SLAM processes are parallel to each other.

Some examples of more tight fusion can also be found in the literature. The work of [107] combined both the laser point cloud data and image feature point data as constraints and conducted a graph optimization with both of these constraints using a specific cost function. Furthermore, an image feature-based loop closure was added to this system to remove accumulation errors.

Inertial SLAM incorporates an inertial measurement unit (IMU) as an assistant sensor. The IMU can be fused with the Camera or Lidar to support pose (position, velocity, attitude) estimation. With an IMU, the attitudes, especially the heading, are observable [108]. The integration of IMU measurements can also improve the motion tracking performance during the gaps of observations. For instance, for a Visual SLAM, illumination change, texture-less area, or motion blur will cause losses of visual tracks [108]. For a Lidar system, the raw Lidar scan data may suffer from skewing caused by high-acceleration motion, such as moving fast or shaking suddenly, resulting in sensing error that is difficult to account for [109]. The work of [110] used an IMU sensor to deal with fast velocity changes and to initialize motion estimates for scan-matching Lidar odometry to support their LOAM system. The high-frequency IMU data between two Lidar scans can be used to de-skew Lidar point clouds and improve their accuracy [109].

The fusion of inertial sensors can be as a simple assistant [111,112] or more tightly coupled [108,113–115]. For the simple assistant case, the IMU is mainly used to provide orientation information, such as to support the system initialization. The IMU is used as prior for the whole system, and the IMU measurements are not used for further optimization. For the tightly coupled case, IMU data is fused with Camera/Lidar states to build up measurement models, and then perform state estimation and feedback to the inertial navigation system to improve navigation performance [116]. Therefore the former method is more efficient than the latter, however, it is less accurate [117]. For the tightly coupled case, a Kalman filter could be used to correct the IMU states, even during GNSS outages [118].

### 2.5. Deep Learning-Based SLAM

Most of the aforementioned SLAM methods are geometric model-based, which build up models of platform motion and the environment based on geometry. These methods have achieved great success in the past decade. However, they still face many challenging issues. For instance, Visual SLAM (VSLAM) is limited under extreme lighting conditions. For large-scale applications, the model-based methods need to deal with large amounts of information, such as features and dynamic obstacles. Recently, deep learning techniques, such as data-driven approaches developed in the computer vision field, have attracted more attention. Many researchers have attempted to apply deep learning methods to SLAM problems.

Most of the current research activities focus on utilizing learning-based methods for VSLAM problems since deep learning techniques have made breakthroughs in the areas of image classification, recognition, object detection, and image segmentation [119]. For instance, deep learning has been successfully applied to the visual odometry (VO) problem, which is an important element of VSLAM. Optical flow estimation is utilized in some learned VO models as inputs [120–124]. The application of learning approaches can be applied in an end-to-end manner without adopting any module in the conventional VO pipeline [125,126]. Wang et al. [125] introduced an end-to-end VO algorithm with deep Recurrent Convolutional Neural Networks (RCNNs) by combining CNNs with the RNNs. With this algorithm, the pose of the camera is directly estimated from raw RGB images, and neither prior knowledge nor parameters are needed to recover the absolute scale [125]. Li et al. [127] proposed an Unsupervised Deep Learning based VO system (UnDeepVO) which is trained with stereo image pairs and then performs both pose estimation and dense depth map estimation with monocular images. Unlike the one proposed by Wang et al. [125], ground truth is not needed for UnDeepVO since it operates in an unsupervised manner.

The learning-based methods can be combined with the VSLAM system to replace or add on an individual or some modules of traditional SLAM, such as image depth estimation [128–130], pose estimation [131–133], and loop closure [134–137], etc., to improve the traditional method. Li et al. [138] proposed a fully unsupervised deep learning-based VSLAM that contains several components, including Mapping-net, Tracking-net, Loop-net, and a graph optimization unit. This DeepSLAM method can achieve accurate pose estimation and is robust in some challenging scenarios, combining the important geometric models and constraints into the network architecture and the loss function.

Sematic perception of the environment and semantic segmentation are current research topics in the computer vision field. They can provide a high-level understanding of the environment and are extremely important for autonomous applications. The rapid development of deep learning can assist in the introduction of semantic information into VSLAM [139] for semantic segmentation [140–142], localization and mapping [143–147], and dynamic object removal [148–151]. Some detailed reviews of deep learning-based VSLAM can be found in [92,139,152–154].

Fusion with an inertial sensor can also benefit from deep learning techniques, especially the RNN, which has an advantage in integrating temporal information and helping to establish consistency between nearby frames [139]. The integration of visual and inertial data with RNN or Long Short-Term Memory (LSTM), a variant of RNN that allows RNN to learn long-term trends [155], has been proven to be more effective and convenient than traditional fusion [156–158]. According to Clark et al. [157], the data-driven approach eliminates the need for manual synchronization of the camera and IMU, and the need for manual calibration between the camera and IMU. It outperforms the traditional fusion method since it is robust to calibration errors and can mitigate sensor drifts. However, to deal with the drift problem, a further extension of the learning-based visual-inertial odometry system to a larger SLAM-like system with loop-closure detection and global relocalization still needs to be investigated.

Compared to the visual-based SLAM, the applications of deep learning techniques for laser scanners or Lidar-based SLAM are still in the early stages and can be considered a

new challenge [159]. Velas et al. [160] used CNN for Lidar odometry estimation by using the IMU sensor to support rotation parameter estimation. The results are competitive with state-of-the-art methods such as LOAM. Li et al. [161] introduced an end-to-end Lidar odometry, LO-Net, which has high efficiency, and high accuracy, and can handle dynamic objects. However, this method is trained with ground truth data, which limits its application to large-scale outdoor scenarios. Li et al. [162] designed a visual-Lidar odometry framework, which is self-supervised, without using any ground truth labels. The results indicate that this VLO method outperforms other current self-supervised visual or Lidar odometry methods, and performs better than fully supervised VOs. Data-driven approaches also make semantic segmentation of Lidar data more accurate and faster, making it suitable for supporting autonomous vehicles [163–165]. Moving objects can be distinguished from static objects by LMNet [166] based on CNNs of 3D Lidar scans. One limitation of some cost-effective 3D Lidar applications for autonomous driving in challenging dynamic environments is its relatively sparse point clouds. In order to overcome this drawback, high-resolution camera images were utilized by Yue et al. [167] to enrich the raw 3D point cloud. ERFNet is employed to segment the image with the aid of sparse Lidar data. Meanwhile, the sparsity invariant CNN (SCNN) is employed to predict the dense point cloud. Then the enriched point clouds can be refined by combining these two outputs using a multi-layer convolutional neural network (MCNN). Finally, Lidar SLAM can be performed with this enriched point cloud. Better target segmentation can be achieved with this Lidar data enrichment neural network method. However, due to the small training dataset, this method did not show improvement in SLAM accuracy with the enriched point cloud when compared to the original sparse point cloud. More training and further investigation of dynamic objects may be needed to satisfy autonomous driving application requirements [167].

The generation of complex deep learning architectures has contributed to achieving more accurate, robust, adaptive, and efficient computer vision solutions, confirming the great potential for their application to SLAM problems. The availability of large-scale datasets is still the key to boosting these applications. Moreover, with no need for ground truth, unsupervised learning is more promising for SLAM applications in autonomous driving. Compared to the traditional SLAM algorithms, data-driven SLAM is still in the development stage, especially for Lidar SLAM. In addition, combining multiple sensing modalities may overcome the shortcomings of individual sensors, for which the learning methods-based integration system still needs further investigation.

## 3. Application of SLAM in Autonomous Driving

Depending on the different characteristics of SLAM techniques, there could be different applications for autonomous driving. One classification of the applications is whether they are offline or online. A map satisfying a high-performance requirement is typically generated offline, such as the High Definition (HD) map [10]. For this kind of 3D point cloud map, an offline map generation process ensures the accuracy and reliability of the map. Such maps can be pre-generated to support the real-time operations of autonomous vehicles.

### 3.1. High Definition Map Generation and Updating

As stated earlier, SLAM can be used to generate digital maps used for autonomous driving, such as the HD map [10]. Due to the stringent requirements, high quality sensors are used. Lidar is one of the core sensors for automated cars as it can generate high-density 3D point clouds. High-end GNSS and INS technology are also used to provide accurate position information. Cameras can provide information that is similar to the information detected by human eyes. The fusion of sensor data and analysis of road information to generate HD maps needs considerable computational power, which is not feasible in current onboard vehicle systems. Therefore the HD map is built-up offline, using techniques such as optimization-based SLAM. The offline map creation can be performed by driving the

road network several times to collect information, and then all the collected perceptive sensor information and position information is processed together to improve the accuracy of the final map. An example of a HD map is shown in Figure 4 [11].

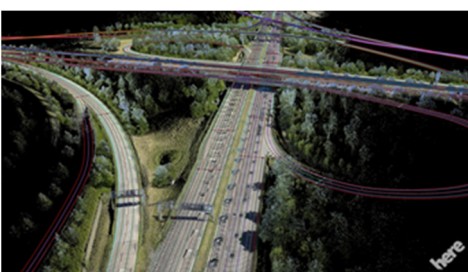

**Figure 4.** An image from a high definition map (https://here.com/) [11].

The road environment and road rules may change, for instance, the speed limit may be reduced due to road work, road infrastructure may be changed due to building development, and so on. Therefore the HD map needs frequent updates. Such updates can utilize the online data collected from any autonomous car. For example, the data is transmitted to central (cloud) computers where the update computations are performed. Other cars can receive such cloud-based updates and make a timely adjustment to driving plans. Jo et al. [168] proposed a SLAM change update (SLAMCU) algorithm, utilizing a Rao–Blackwellized PF approach for online vehicle position and (new) map state estimation. In the work of [169], a new feature layer of HD maps can be generated using Graph SLAM when a vehicle is temporarily stopped or in a parking lot. The new feature layer from one vehicle can then be uploaded to the map cloud and integrated with that from other vehicles into a new feature layer in the map cloud, thus enabling more precise and robust vehicle localization. In the work of Zhang et al. [170], real-time semantic segmentation and Visual SLAM were combined to generate semantic point cloud data of the road environment, which was then matched with a pre-constructed HD map to confirm map elements that have not changed, and generate new elements when appearing, thus facilitating crowdsource updates of HD maps.

### 3.2. Small Local Map Generation

SLAM can also be used for small local areas. One example is within parking areas. The driving speed in a parking lot is low, therefore the vision technique will be more robust than in other high-speed driving scenarios. The parking area could be unknown (public parking lot or garage), or known (home zone)–both cases can benefit from SLAM. Since SLAM can be used without GNSS signals, it is suitable for vehicles in indoor or underground parking areas, using just the perceptive sensor and odometry measurements (velocity, turn angle) or IMU measurements. For unknown public parking areas, the position of the car and the obstacles, such as pillars, sidewalls, etc., can be estimated at the same time, guiding the parking system. For home zone parking, the pre-generated map and a frequent parking trajectory can be stored within the automated vehicle system. Each time the car returns home, re-localization using the stored map can be carried out by matching detected features with the map. The frequent trajectory could be used for the planning and controlling steps.

An approach that utilizes multi-level surface (MLS) maps to locate the vehicle, and to calculate and plan the vehicle path within indoor parking areas was proposed in [171]. In this study, graph-based SLAM was used for mapping, and the MLS map is then used to plan a global path from the start to the destination, and to robustly localize the vehicle with laser range measurements. In the work of [172], a grid map and an EKF SLAM algorithm were used with W-band radar for autonomous back-in parking. In this work, an efficient EKF SLAM algorithm was proposed to enable real-time processing. In [173], the authors proposed an around-view monitor (AVM)/ Lidar sensor fusion method to recognize the parking lane and to provide rapid loop closing performance. The above

studies have demonstrated that both filter-based SLAM and optimization-based SLAM can be used to support efficient and accurate vehicle parking assistance (local area mapping and localization), even without GNSS. In the work of Qin et al. [174], pose graph optimization is performed so as to achieve an optimized trajectory and a global map of a parking lot, with semantic features such as guide signs, parking lines, and speed bumps. These kinds of features are more (long-term) stable and robust than traditional geometrical features, especially in underground parking environments. An EKF was then used to complete the localization system for autonomous driving.

### 3.3. Localization within the Existing Map

In map-based localization, a matching method is used to match "live" data with map information, using methods such as Iterative Closest Point (ICP), Normal Distribution Transform (NDT), and others [10,175]. These algorithms can be linked to the SLAM problem since SLAM executes loop closing and re-localization using similar methods. For a SLAM problem, the ability to recognize a previously mapped object or feature and to relocate the vehicle within the environment is essential for correcting the maps [13]. Therefore, the reuse of a pre-generated map to localize the vehicle can be considered an extension of a SLAM algorithm. In other words, the pre-generated and stored map can be treated as a type of "sensor" to support localization.

However, matching live data with a large-scale pre-prepared map requires substantial computational resources. Hence, some methods have been proposed to increase computational efficiency. One method is to first narrow down the possible matching area from the map with position estimated from GNSS or GNSS/INS, and then carry out detailed matching of the detected features with the map [176].

Due to the current limited installation of Lidar systems in commercial vehicles (high price of sensor and high power consumption), localization of a vehicle with a low-cost sensor (e.g., vision sensor) in a pre-generated HD map is of considerable practical interest. For instance, the work in [177] located a vehicle within a dense Lidar-generated map using vision data and demonstrated that a similar order of magnitude error rate can be achieved to traditional Lidar localization but with several orders of magnitude cheaper sensor technology. Schreiber et al. [178] proposed to first generate a highly accurate map with road markings and curb information using a high-precision GNSS unit, Velodyne laser scanner, and cameras. Then during the localization process, a stereo camera system was used to detect road information and match it with the pre-generated map to achieve lane-level real-time localization. Jeong et al. [179] utilized road markings obtained from camera images for global localization. A sub-map that contained road information, such as 3D road marking points, was generated and utilized to recognize a revisited place and to support accurate loop detection. The pose graph-based approach was then used to eliminate the drift. Qin et al. [146] proposed a semantic localization system to provide a light-weight localization solution for low-cost cars. In this work, a local semantic map was generated by combining the CNN-based semantic segmentation results and the optimized trajectory after pose graph optimization. A compacted global map was then generated (or updated) in the cloud server for further end-user localization based on the ICP method and within an EKF framework. The average size of the semantic map was 36 kb/km. This proposed camera-based localization framework is reliable and practical for autonomous driving.

In addition to the aforementioned applications, moving objects within the road environment will cause a drift of perception, localization, and mapping for autonomous driving. SLAM can be used to address the problem of DATMO (detection and tracking of moving objects) [15] because one of the assumptions of SLAM is that the detected features are stationary. As the static parts of the environment are localized and mapped by SLAM, the dynamic parts can be concurrently detected and tracked. Some approaches have dealt with dynamic obstacles [180–182].

## 4. Challenges of Applying SLAM for Autonomous Driving and Suggested Solutions

*4.1. Ensuring High Accuracy and High Efficiency*

Localization and mapping for automated vehicles need to be accurate and robust to any changes in the environment and executed with high efficiency. With rapidly developing sensor technology, the combination of different sensors can compensate for the limitations of a particular sensor. Examples include GNSS/INS + Lidar/Camera SLAM, Radar SLAM, and some others. There is considerable research and development associated with low-cost and/or miniaturized Lidar sensors. New Lidar sensor concepts promise a significant reduction in the cost of Lidar systems, with the potential for real-time implementation in future autonomous vehicles. For instance, RoboSense has unveiled a new $200 Lidar sensor combining MEMS sensors and an AI-based deep-learning algorithm to support high-performance autonomous driving applications [183].

Choosing a SLAM approach should take into consideration different application scenarios with different level of requirements. Optimization-based SLAM can provide more accurate and robust estimation, however, it is more suitable for offline estimation. EKF SLAM suffers from the quadratic increase in the number of state variables, which restricts its online application in large-scale environments. Although high-resolution map generation can be offline, real-time, or near-real-time, solutions are essential for map updating and map-based localization applications.

Any change in the road environment should be quickly updated on the map and transmitted to other road users. Emerging 5G wireless technology can make the communication between vehicle-to-vehicle (V2V), vehicle-to-infrastructure (V2I), and vehicle-to-cloud more reliable and with higher throughput [14].

*4.2. Representing the Environment*

There are different types of maps that can be used to represent the road environment. Three major types of maps in the robotic field for SLAM applications are occupancy grid maps, feature-based maps, and topological maps [184]. They are also applicable to road environments. Each of them has its own advantages and limitations for autonomous driving applications. The grid map divides the environments into many fixed-size cells, and each cell contains its own unique property, such as whether the grid is occupied, free or unknown [185,186]. The obstacle occupancy information can be directly fed to the planning algorithms. This kind of map can be merged easily and has flexibility in incorporating data from numerous types of sensors [184]. Mentasti and Matteucci [185] proposed an occupancy grid creation method that utilized data from all the available sensors on an autonomous vehicle, including Lidar, Camera, Laser, and Radar. The grid map also shows the potential for detecting moving objects [187]. Mutz et al. [188] compare the performance of mapping and localization with different grid maps, including occupancy, reflectivity, color, and semantic grid maps, for self-driving car applications in diverse driving environments, including under challenging conditions. GraphSLAM was used for mapping, while localization was based on particle filter solutions. According to their results, the occupancy showed more accurate localization results, followed by the reflectivity grid map. Semantic grid maps kept the position tracking without losses in most scenarios, however with bigger errors than the first two map approaches. Colorized grid maps were most inconsistent and inaccurate for use in localization, which may be due to the influence of illumination conditions. One shortcoming that limits the occupancy grip map for large-scale autonomous driving is its dense representation, which needs big storage space and high computation power [189]. Thus Li [186] suggested applying this technique for real-time local mapping with a controlled size instead of for global mapping.

The feature-based map is a popular map type for autonomous driving. It represents the map with a group of features extracted from sensor data. For outdoor road environments, the typical features are traffic lanes, kerbs, road markings and signs, buildings, trees, etc. For indoor areas, especially in parking areas, the features are mainly the parking lane, sidewalls, etc. These features can be represented by points, lines, and planes, tagged

with coordinate information. The point feature represents the environment as dense point clouds. The high-density point cloud maps generated using Lidar and/or vision sensors can provide abundant features and 3D structure information of the area surrounding the vehicle. However, the transmission, updating, and processing of this volume of data is burdensome for complex road environments. The sparser line and plane features are suitable for structured environments, such as indoor environments, urban areas, or highways, with clear markings. These features are more sophisticated than the point features, with lower memory requirements [186], and are less susceptible to noise [189]. Im et al. [173] proposed a parking line-based SLAM approach which extracted and analyzed parking line features to achieve rapid loop closure and accurate localization in a parking area. Javanmardi et al. [190] generated a city road map with 2D lines and 3D planes to represent the buildings and the ground along the road. However, for autonomous driving, the application environment is variable. A specific landmark-based algorithm may not be suitable for other driving scenarios. Furthermore, in some rural areas, the road may be unpaved and there is no road lane marking. Thus the related feature-based map approaches may not be feasible due to the lack of road markings and irregular road curve geometry [191].

The topological map represents the environment with a series of nodes and edges. The nodes indicate the important objects, such as corners, intersections, and feature points; while the edges denote the topological relationships between them [192,193]. One typical topological map is OpenStreetMAP (OSM) [194] which contains the coordinates of features as well as road properties such as road direction, lane numbers, etc. This kind of map significantly reduces the storage and computational requirements. However, it loses some useful information about the nature and structure of the actual environment [184]. Thus, some approaches combine topological maps with other types of maps. Bernuy and Ruiz-del-Solar [195] proposed the use of a topological map based on semantic information to provide robust and efficient mapping and localization solutions for large-scale outdoor scenes for autonomous vehicles and ADAS systems. According to Bernuy and Ruiz-del-Solar [195], the graph-based topological semantic mapping method was suitable for large-scale driving tasks on highways, rural roads, and city areas, with less computational expense than metrics maps. Bender et al. [196] introduced a highly detailed map, Lanelets, which combines both geometrical and topological representations, and includes information on traffic regulations and speed limits.

The semantic map is becoming increasingly important in autonomous fields as it contains semantic information that allows the robot or vehicle to better understand the environment, and to complete higher-lever tasks, such as human-robot interaction. For outdoor applications, the labeled objects could be statistic background (e.g., 'building', 'tree', 'traffic sign') or dynamic entities (e.g., 'vehicle', 'pedestrian'). Therefore, this kind of map can facilitate complex tasks for autonomous vehicles, such as planning and navigation [195,197]. Associating semantic concepts with geometric entities has become a popular research topic and semantic SLAM approaches have been investigated that combine geometric and semantic information [139,143,149]. The semantic SLAM approaches can contribute to making localization and mapping more robust [174], to supporting re-localization at revisited areas [143], and very importantly, to tracking moving objects detected in dynamic environments [149,151,198]. One critical problem faced by semantic map generation and utilization is that some modules within them, such as semantic segmentation, are very computationally demanding, which makes them unsuitable for real-time applications [199], especially for large-scale outdoor scenarios. Thus, some investigations seek to solve this problem. Ros et al. [199] proposed an offline-online strategy that generates a dense 3D semantic map offline without sacrificing accuracy. Afterward, real-time self-localization can be performed by matching the current view to the 3D map, and the related geometry and semantics can be retrieved accordingly. Meanwhile, the new dynamic objects can be detected online to support instantaneous motion planning. With the advent of deep learning, the efficiency and reliability of semantic segmentation and semantic SLAM have been vastly improved [147,200–203]. However, as previously mentioned, when applying

deep learning-based semantic SLAM to autonomous driving, there are still some challenges, such as the need for large amounts of training data, or the lack of ground truth that makes unsupervised learning methods necessary.

The different map representations are essential to support a highly automated vehicle operating in a challenging and complex road environment. Therefore, a detailed digital map, such as the HD map, which contains different layers of data, has been increasingly adopted. In addition to the most basic 3D point cloud map layer, the HD map may also contain layers with information on road topology, geometry, occupancy, lane features, road furniture, road regulation, real-time knowledge, and more. The storing, updating, and utilizing of such dense data without losing accuracy is a challenge. Some researchers have proposed the concept of "Road DNA" to represent the road environment and to deal with the Big Data problem [12,204]. Road DNA converts a 3D point cloud road pattern into a compressed, 2D view of the roadway without losing details [12], with the objective to reduce processing requirements.

### 4.3. Issue of Estimation Drifts

SLAM estimation drifts may be caused by accumulated linearization error, the presence of dynamic obstacles, noisy sensor data, wrong data association, etc.

In most SLAM algorithms, nonlinear models are used to represent the vehicle motion pattern and the environment. EKF SLAM suffers from a divergence problem due to the accumulation of linearization errors. Biases may occur when linearization is performed using values of state variables that are far from their true values. For optimization-based SLAM, a poor initial guess of variables will lead to poor convergence performance. Rotation may be the cause of nonlinearity and has a strong impact on the divergence of estimation [205,206], thus the accumulated vehicle orientation error will cause the inconsistency of the SLAM problem. One solution to the linearization challenge is the Linear SLAM algorithm proposed in [205], which modifies the relative state vector and carries out "map joining". Sub-map joining, which involves solving a linear least squares problem and performing nonlinear coordinate transformations, does not require an initial guess or iteration. In the work of [207], a robocentric local map sequencing approach was presented which can bound location uncertainty within each local map and improve the linearization accuracy with sensor uncertainty level constraints. Many variants of the classical EKF-SLAM have been proposed to overcome the divergence of the filter. The study of [208] demonstrated that the Unscented SLAM can improve the online consistency for large-scale outdoor applications. Huang et al. [209] proposed two alternatives for EKF-SLAM, Observability Constrained EKF, and First-Estimates Jacobian EKF, both of which significantly outperform the EKF in terms of accuracy and consistency. A linear time varying (LTV) Kalman filtering was introduced in [210] which avoids linearization error by creating virtual measurements. Some nonparametric approaches which are mainly based on the PF, such as fastSLAM [28], Unscented fastSLAM [211–214], show better performance than the EKF-SLAM.

For the nonlinear optimization-based SLAM approach, computing a good initial guess (solving the initialization problem), will lead to faster convergence and reduce the risk of convergence to local minima. Olson et al. [215] presented a fast iterative algorithm for optimizing pose graphs using a variant of Stochastic Gradient Descent (SGD), which is robust against local minima and converges quickly even with a bad initial guess. Then in the work of [50], an extension of Olson's algorithm was proposed which uses a tree-based parameterization for the nodes in the graph. This algorithm was demonstrated to be more efficient than Olson's and robust to the initial configuration. An approximation solution for 2D pose-graphs, called Linear Approximation for a pose Graph Optimization (LAGO), can be used as an exact solution or for bootstrapping nonlinear techniques [216,217]. This method first solves a linear estimation problem to obtain the suboptimal orientation estimate, and then uses it to estimate the relative position measurements in the global reference frame. Finally, the position and orientation solution is obtained by solving another linear estimation problem. This solution can then be treated as an initial guess

for a Gauss-Newton iteration. This method can provide a good initial guess, however, it is limited to 2D pose-graph, and is sensitive to noisy measurements. An algorithm with more complex initialization was proposed in [218] that uses the M-estimator, in particular the Cauchy function, as a bootstrapping technique. Similar to approaches that use the M-estimator to make estimation robust to outliers, the M-estimator proved to also be robust to a bad initial guess. In contrast to LAGO and TORO, this method can be applied to different variants of SLAM (pose-graphs and feature-based) in both 2D and 3D [218]. Carlone et al. [219] surveyed different 3D rotation estimation techniques and demonstrated the importance of good rotation estimate to bootstrap iteration pose graph solvers. More recent research presented a heuristic method called Multi-Ancestor Spatial Approximation Tree (MASAT), which has low complexity and is computationally efficient without needing a preliminary optimization step [220]. This method is still for the pose graph. Other studies seek to obtain a good initial guess by introducing inertial measurements to support initialization [221,222] or conducting parameter calibration [223–225].

Dynamic objects such as pedestrians, bicycles, other vehicles, etc., may cause estimation drifts since the system may wrongly identify them as static road entities. There are some methods to avoid this. Probabilistic Maps that use probabilistic infrared intensity values have been proposed in [226]. In this study, GNSS/INS and a 64-beam Lidar sensor were combined to achieve robust position RMS errors of 9 cm in dynamic environments. However, this system suffers from high costs and a high computational burden. The 3D Object Tracker [227] can be used to track moving objects in Visual SLAM methods. Another algorithm proposed in [228] uses Canny's edge detector to find dominant edges in the vertical direction of a tree trunk and to select these tree trunks as typical salient features. Deep learning methods are increasingly investigated to deal with the dynamic environment as aforementioned [148–151,166,198].

Another source of drifts is the outlier within the sensor observations. Each sensor has its own error sources. For example, in the case of a camera, the fuzzy image due to high speed and poor light conditions may cause incorrect identification of landmarks. Lidar sensors are sensitive to weather conditions (such as rainfall), and large changes in the road environment. GNSS may suffer from signal blockage. FDI (Fault Detection and Isolation system) techniques can be used to detect measurement outliers and reject the influence of these outliers on positioning and localization [229].

The aforementioned SLAM error sources may also result in incorrect data association, which is an important process to associate measurement(s) to a specific landmark. Wrong data association may happen due to not only the noisy sensor data, inconsistency, wrong detection of dynamic objects, etc., but also to some specific road environments. For instance, the highway environment is sometimes visually repetitive and contains many similar features, which makes it difficult to recognize a previously explored area.

Some researchers avoid the challenge of wrong data association directly at the frontend step of SLAM by using RANSAC [230], which is commonly used in Visual SLAM to reject outliers. In [231], the authors proposed a middle layer, referred to as Graph-Tinker (GTK), that can detect and remove false-positive loop closures. Artificial loop closures are then injected into the pose graph when using an Extended Rauch–Tung–Striebel smoother framework.

The data association challenge can also be addressed at the backend step since there is still a chance that outliers are not totally eliminated. The concept of Switchable Constraints (SC) was introduced in [232], such that a switchable variable is introduced into each loop closure constraint. Once a constraint is considered as an outlier, it can be turned off during optimization. In [233], the authors introduced an algorithm known as Realizing, Reversing, and Recovering (RRR), which is a consistency-based loop closure verification method. More recently, Carlone et al. [234] used $\ell1\_relaxation$ to select "reliable" measurements, and Carlone and Calafiore [235] use convex relaxations to solve the nonconvex problem without the need for an initial guess of unknown poses. The potential causes of SLAM drifts and the corresponding suggested solutions are summarized in Table 2.

**Table 2.** Potential causes of SLAM drifts and solutions.

| SLAM Drift | Possible Solutions |
|---|---|
| Linearization error | • Variants of EKF-SLAM [208–210];<br>• Nonparametric approaches [28,211–214];<br>• Local map joining [205,207];<br>• Gradient Descent based optimization scheme [50,215];<br>• Improved initialization method [216–218,220];<br>• Aiding with inertial measurements [221–225]. |
| Sensor outliers | • Fault Detection and Isolation [229];<br>• Sensor fusion to compensate for different sensor errors [236,237]. |
| Dynamic objects | • Probabilistic maps [226];<br>• 3D Object Tracker [227];<br>• Salient feature detection [228];<br>• Deep learning-based methods [148–151,166,198]. |
| Wrong data association | • RANSAC [230];<br>• Graph-Tinker [231];<br>• Switchable Constraints [232];<br>• RRR [233];<br>• $\ell1$_relaxation [234], convex relaxations [235]. |

*4.4. Lack of Quality Control*

The quantitative evaluation of the SLAM algorithms is another important challenge. There are some criteria to evaluate SLAM algorithms, such as their accuracy, scalability, availability, recovery (which is the ability to localize the vehicle inside a large-scale map), and updatability. Quantitative analysis of the performance of SLAM algorithms is essential since they can provide numerical evaluation and a basis for comparison of different SLAM algorithms.

Estimation accuracy is a widely used quality analysis metric, but it can be difficult in practice for autonomous driving. Most approaches evaluate the performance of SLAM algorithms by comparing the results to ground truth using, for example, an accurate map. However, a suitable ground truth map is seldom available. Sometimes the estimated map is evaluated by overlaying it onto the floor plan and searching for differences [238], which is harder for outdoor applications and needs human operator intervention [239]. Two popular accuracy metrics, relative pose error (RPE) and absolute trajectory error (ATE), were proposed by Sturm et al. [240] which evaluate a Visual SLAM system by comparing the estimated camera motion against the true trajectory, instead of doing complex map comparison. The RPE measures the local accuracy of trajectory over a fixed time interval, while ATE compares the absolute distances between the estimated and the ground truth trajectory and thus estimates the global consistency. These two trajectories should first be aligned using the Horn method [240]. According to [240], the RPE considers both translational and rotational errors, while the ATE only considers the translational error. These metrics have been widely used by the SLAM community for evaluating and comparing different SLAM approaches. However, similar to maps, the precise location of the vehicle trajectory on the actual road surface may not always be available. In [239], the authors proposed a framework for analyzing the accuracy of SLAM by measuring the error of the corrected trajectory. Root Mean Square Error (RMSE) of vehicle poses is normally used to indicate the accuracy of the SLAM trajectory estimation result. Another widely used quality analysis method is the Chi-squared ($\chi^2$) test. According to [241], the $\chi^2$ test is a statistic test to quantify the quality of the provided covariance matrices for landmark measurements and odometry error. When the minimum $\chi^2$ error is nearly equal to the difference of the dimension of the measurement vector and the size of the state vector, the measure would be considered as being of good quality [241].

Some researchers [242–244] have considered the consistency of their SLAM algorithms. According to [242], the major reason for SLAM inconsistency is the accumulated error caused by the incorrect odometry model and inaccurate linearization of the SLAM nonlinear functions. When the estimation error is beyond the uncertainty, it can be assumed that the estimation results are inconsistent. EKF-SLAM suffers from such an inconsistency problem unless the Jacobians of observation/odometry functions are evaluated around the true system state. In [30] and [245], the consistency of fastSLAM and EKF-SLAM algorithms was quantitatively determined using the measure indicator normalized estimation error squared (NEES). In [246], observability properties of the filter's error state model were analyzed to investigate the fundamental causes of the inconsistency of EKF-SLAM. In the work of [247], the consistency of an incremental graph SLAM was checked by applying a $\chi^2$ test to the weighted sum of measurement residuals. Whether inconsistency can be tolerated ultimately depends on the application of the SLAM results [19].

The reliability of the output of the localization, mapping, and navigation system should also be checked. However, few studies have been made on the quantitative analysis of the reliability of SLAM. Some reliability studies for other localization systems (such as GNSS, GNSS/INS) can be used as a reference to guide the SLAM community. System reliability can be considered as having two components: internal reliability and external reliability. The former identifies the ability of the system to detect faults, which is quantified by the Minimal Detectable Bias (MDB), and is indicated by the lower bound for detectable faults. The latter estimates the influence of undetected faults on the final solution [175,248–251]. When the MDB value is low, the system is more reliable. Similarly, the reliability of the SLAM system feature observation model and vehicle motion model can also be evaluated with these approaches.

Integrity is very important, as it is an indicator of the "trustworthiness" of the information supplied by the localization system, and can provide timely warning of the risks caused by inaccuracy [252]. Integrity measures are used to quantify the requirements for localization safety. The concept was first established in aviation and is also applicable to land vehicle localization [253]. Due to the strict safety requirement of autonomous driving, there is increasing attention to integrity by autonomous driving researchers. The localization and navigation of a self-driving car are based on the use of multiple sensors, therefore the traditional integrity analysis methods for GNSS should be extended. Fault detection and isolation (FDI) is one of the most popular alert generation approaches for GNSS-based localization [229,254–256].

## 5. Lidar/GNSS/INS Based Mapping and Localization: A Case Study

The Lidar-based Simultaneous Localization and Mapping (SLAM) technology approach is widely studied and used in the robotics field because Lidar can generate a very dense 3D point cloud with a fast sensing rate and high accuracy. Normally the SLAM system experiences estimation error which increases with the travel distance, thus it needs "loop closure" to correct the errors. However, the closed loop is hard to achieve in some large-scale outdoor applications of autonomous driving, such as driving on a highway, or a complex trajectory in urban areas. Furthermore, the Lidar-only SLAM will only provide the relative localization information. Therefore, the combination of GNSS/INS with Lidar SLAM will effectively reduce the dependence on loop closures and provide absolute positioning information.

Furthermore, a Lidar system can also support localization using existed HD maps when GNSS signals are not available. A modernized SLAM procedure that combines Lidar, GNSS, and INS is tested here. This procedure contains two parts: Lidar/GNSS/INS-based offline mapping part, and Lidar/HD map-based online localization and mapping part.

### 5.1. Experiment Setup

Land vehicle tests were conducted in some urban areas of Sydney, Australia, to test the proposed Lidar/GNSS/INS multi-sensor system. The vehicle was equipped with a VLP-16

LiDAR sensor, a tactical-grade IMU sensor, and two GNSS antennas from PolyExplore, Inc., San Jose, CA, USA (Figure 5). The second antenna can be used to provide a dual-antenna-aided heading update for the online localization system. The sampling rate of the Lidar was 10-Hz, the sample rate of GNSS was 1-Hz, and for the IMU it was 100-Hz.

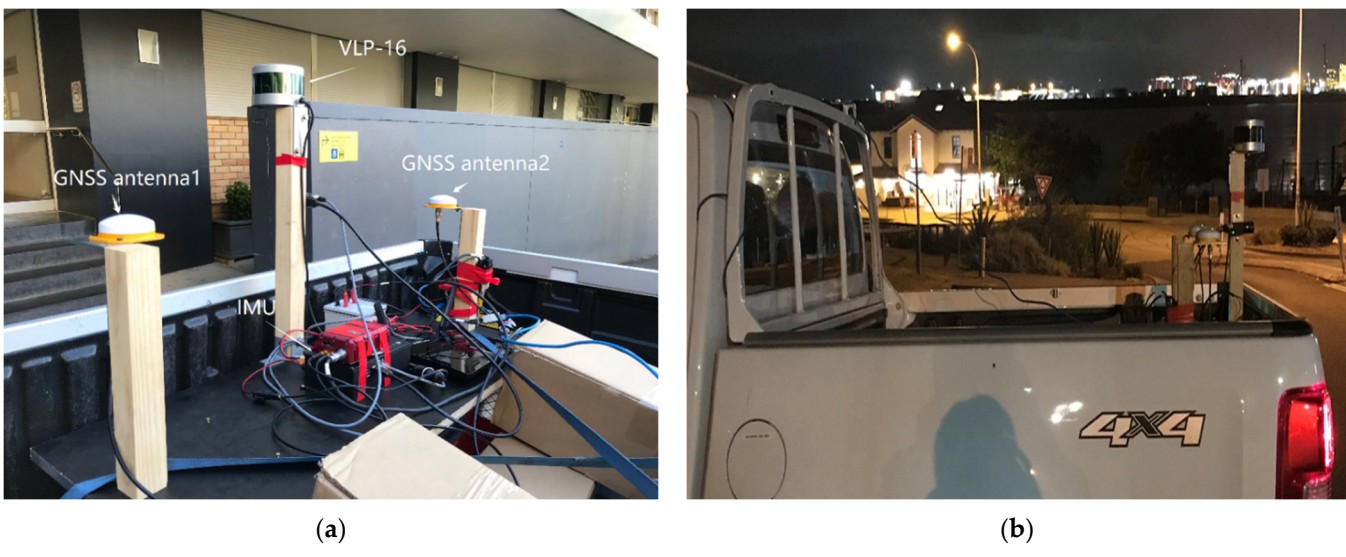

(**a**)  (**b**)

**Figure 5.** Experimental platform: (**a**) The multi-sensor system, (**b**) side view of the system installed within a vehicle.

The trajectory of the road test is shown in Figure 6a. The vehicle was driven from the campus of the University of New South Wales (UNSW) in Kensington to La Perouse (Section A), and then back to UNSW (Section B). In this study, the forward journey (from UNSW to La Perouse) was used to a produce high precision 3D point cloud map of the road, and the backward journey (from La Perouse to UNSW) was used to test the performance of the Lidar/3D point cloud map-based localization method.

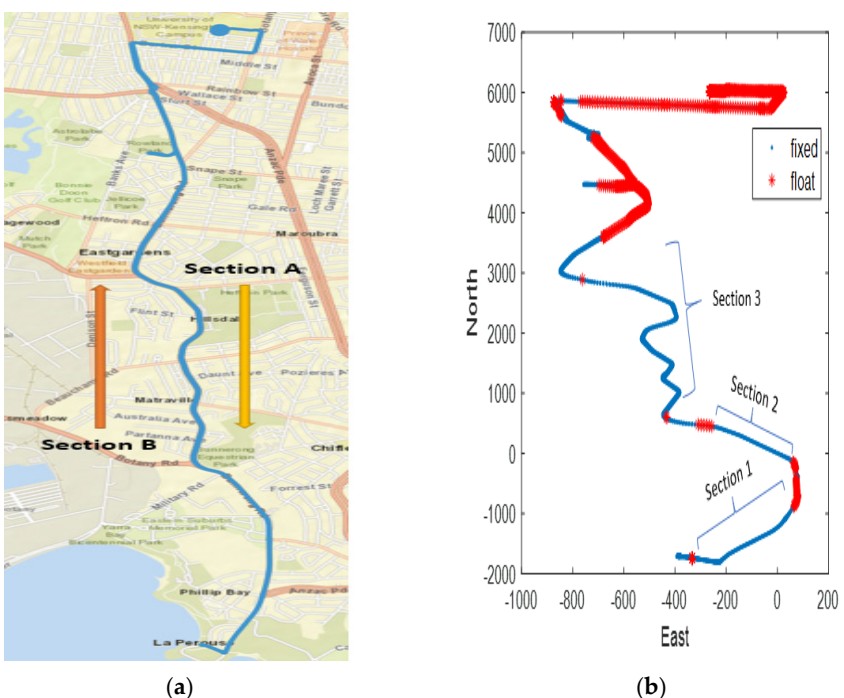

(**a**)  (**b**)

**Figure 6.** (**a**) The road test trajectory (in blue) on Google Maps; (**b**) GNSS/INS localization of the whole trajectory with RTK positioning status in a local coordinate system.

In order to conduct a quantitative analysis of the localization performance, three sections of the trajectory were selected (Figure 6b). For each of the selected sections on the driving trajectory, the GNSS-RTK status for the forward journey and backward journey was "integer-ambiguity fixed". Hence, the offline mapping results are expected to be accurate at about the 5 cm level. For the backward journey (from La Perouse to UNSW), the selected sections will have accurate GNSS/INS positioning results as a reference to evaluate the performance of the Lidar/3D point cloud map-based localization method.

### 5.2. Lidar/GNSS/INS Mapping

The acquired dataset of the forward journey (from UNSW to La Perouse) was used to generate a georeferenced point cloud map of the road environment. The georeferenced map was generated using Lidar odometry frame-to-frame matching and GNSS/INS positioning/attitude. Figure 7 shows an overview of the offline mapping system architecture.

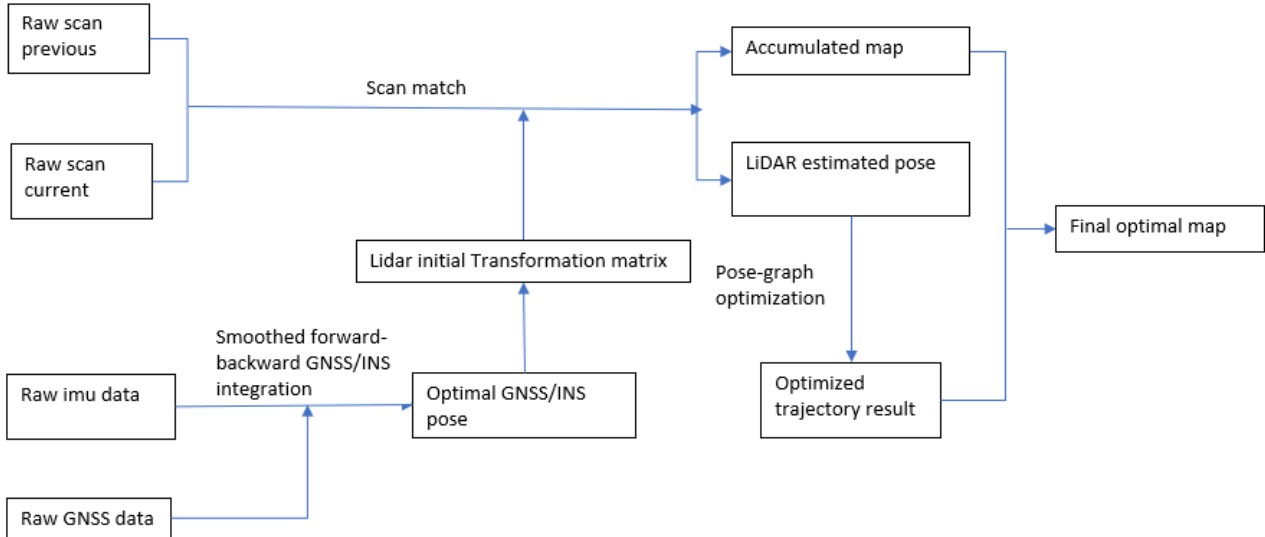

**Figure 7.** Overview of the Lidar/GNSS/INS mapping system architecture.

The GNSS/INS system can provide the geodetic positioning and attitude information. Since this map generation was performed offline, an optimal GNSS/INS trajectory can be obtained. The GNSS/INS-derived position and attitude results were used as initial values for the frame-by-frame matching to transfer the newly merged point cloud to the referenced frame. In this way, the point cloud can be georeferenced. When GNSS results are unavailable the inertial navigation 6-DOF pose results can be used to generate the initial transformation before the GNSS signals are reacquired. When conducting Lidar odometry, each current frame was matched to the previous frame with Normal Distributions Transform (NDT) scan matching algorithm, with the initial transformation information provided by GNSS/INS. The point clouds were firstly pre-processed to remove the ground plane point (Figure 8), before matching by NDT to improve the accuracy of registration.

Figure 9 shows two scan views before being matched. It appears that these two scan views have slight differences in features. The matched point cloud from the two scan views can be generated (Figure 10).

By conducting Lidar odometry sequentially with all the available Lidar scans, the newly matched point cloud can be merged with the previously generated point cloud maps, and the accumulated map of the whole trajectory can be obtained and georeferenced, as shown in Figure 11.

By enlarging Figure 11, details of the road map can be seen, and its corresponding real-world road view can be compared with Google Earth images (since this map is georef-

erenced). Figure 12 shows a comparison of one zoomed-in section of this generated map and the corresponding view in Google Earth.

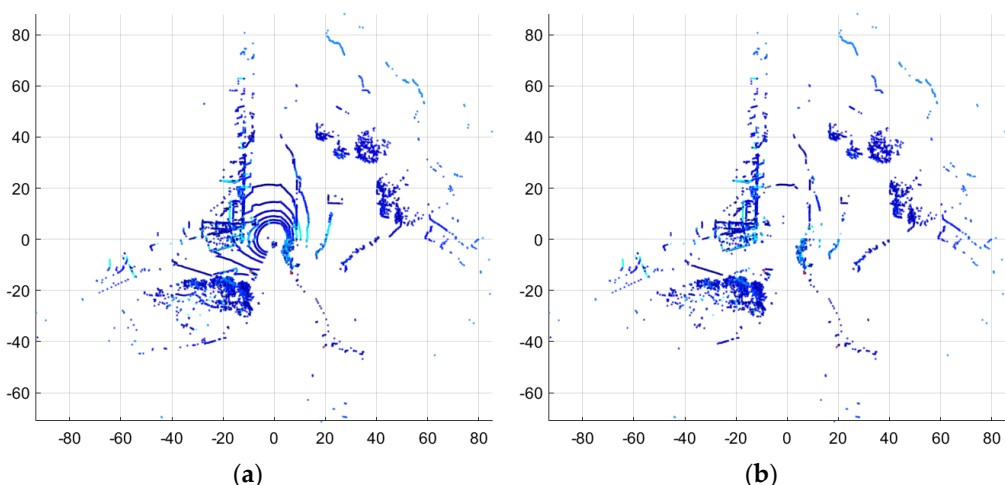

**Figure 8.** Scan view of a Lidar scan frame (**a**) the original scan view; (**b**) the view after pre-processing.

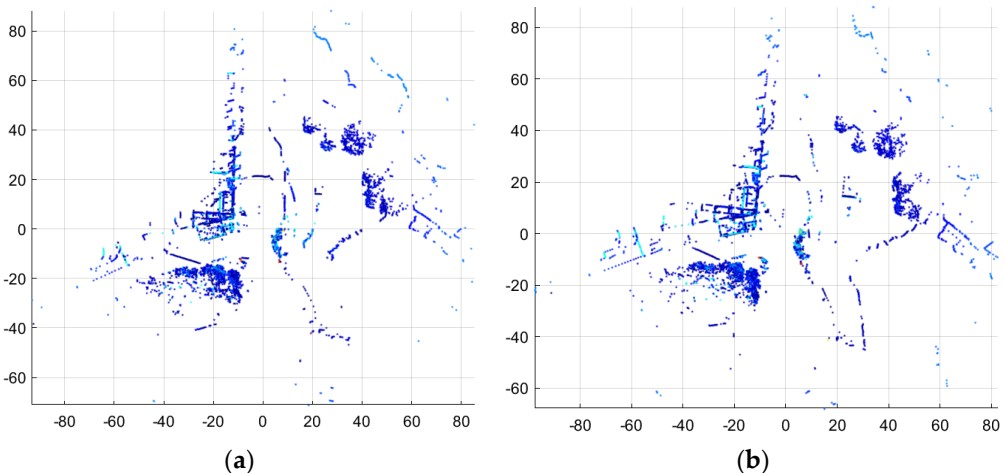

**Figure 9.** Scan views of two sequenced Lidar scan frames ((**a**) previous scan frame; (**b**) current scan frame) for scan matching.

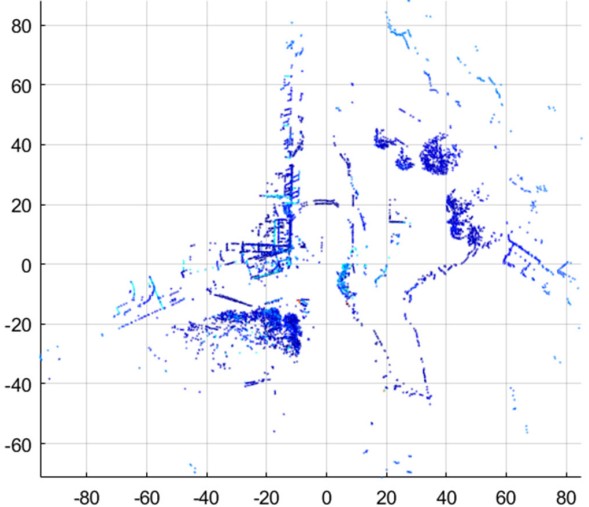

**Figure 10.** Generated map point cloud after matching two sequenced Lidar scan frames.

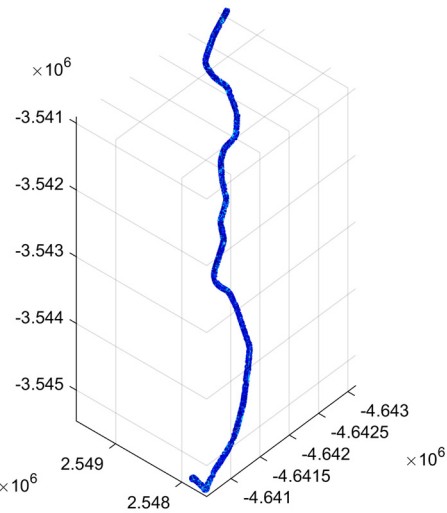

**Figure 11.** Global georeferenced road map from UNSW to La Perouse (frame: ECEF, unit: meter) from the 3D point cloud-based map.

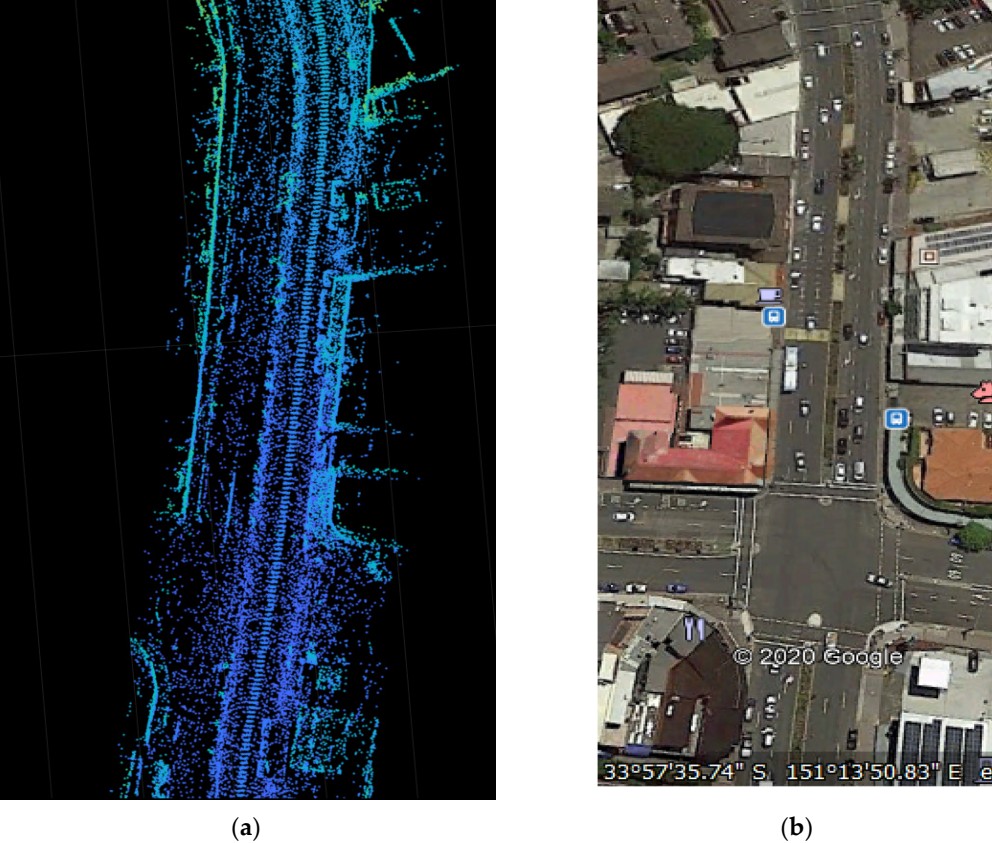

(**a**)　　　　　　　　　　　　　　　　(**b**)

**Figure 12.** A section of (**a**) generated map, and (**b**) the Google Earth view for the same location.

This generated map shows a good structure of the road environment, including the road edge, buildings, trees, and parked vehicles along the road.

Three control points with known coordinates are placed around the UNSW Scientia Lawn. These control points can be used to evaluate the accuracy of the generated point cloud. By comparing the coordinates of the identified control points within the map to the real known position, it is found the difference at each axis X, Y, and Z is around 2–8 cm. Therefore, the offline-generated map accuracy is considered to be 5 cm.

### 5.3. Localization with Lidar Scans and the GeoReferenced 3D Point Cloud Map Matching

The georeferenced 3D point cloud map produced from the data of Section A (the forward journey from UNSW to La Perouse) can then be used to support the Lidar-based localization for Section B (the backward journey from La Perouse to UNSW) by matching the Lidar scans to the map. The procedure of the online Lidar/3D map matching-based localization method is shown in Figure 13. An INS is used to support the Lidar/3D map-based online localization. In order to show the performance of different fusion levels, two fusion methods were investigated. The first method simply utilizes the IMU as an assistant sensor that directly uses the INS solution as initial information for scan/map matching. The second fusion method is a tightly coupled one that not only uses the INS solution to support matching, but also contains an EKF-based error state update step that enhances the inertial navigation performance.

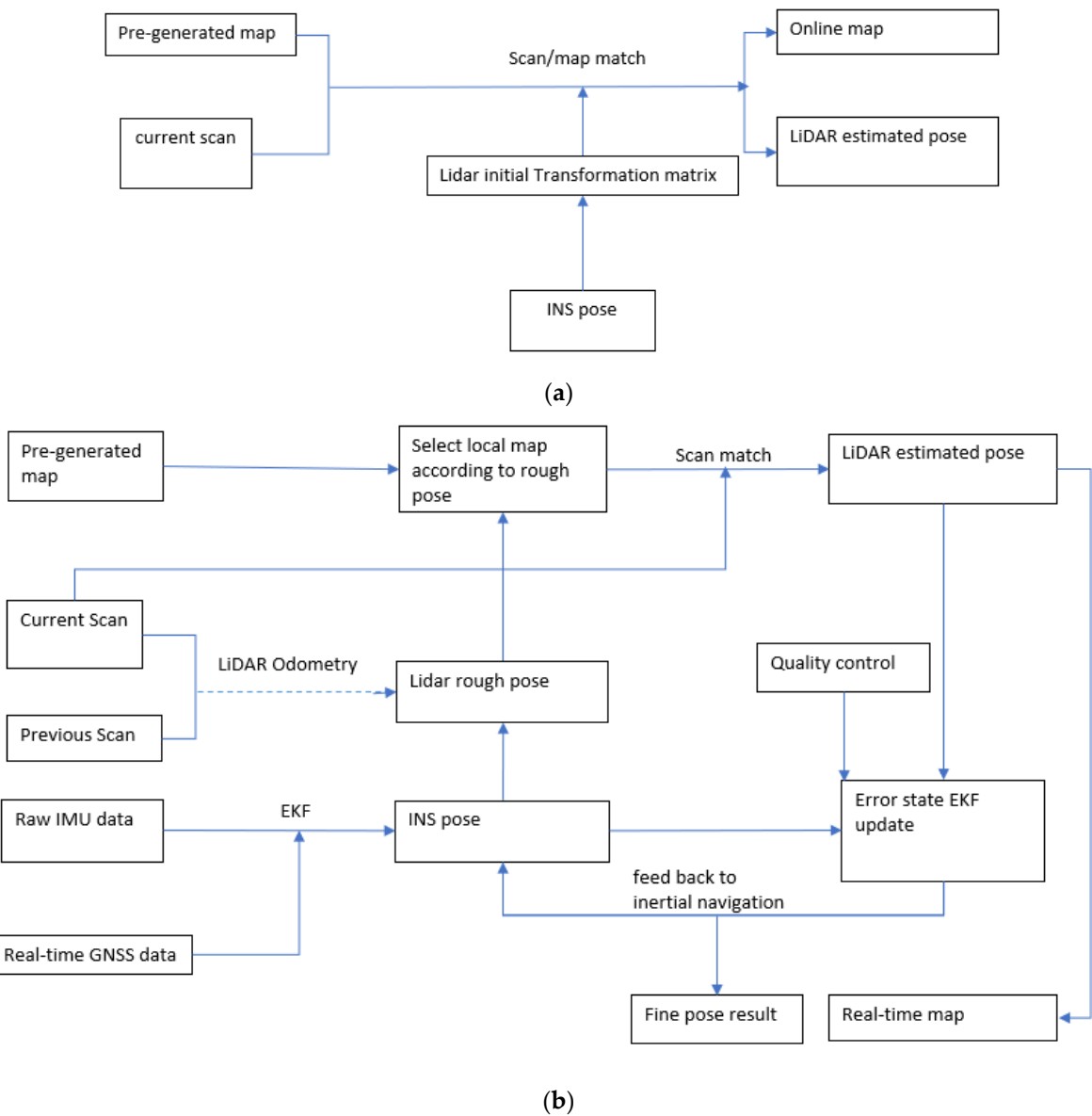

**Figure 13.** Overview of the proposed Lidar/3D map matching based localization system architecture. (**a**) Method 1: fusing IMU as an assistant sensor; (**b**) Method 2: fusing IMU using the EKF-based tightly coupled method.

Method 2 consists of two parts: scan matching and EKF fusing. Firstly, if the inertial navigation information is not available, the frame-to-frame Lidar odometry can be used

to support localization. After initializing the error-state EKF, the estimated pose from the inertial navigation will provide a rough pose for the current Lidar scan frame, and the Lidar odometry can be shut down to lower the computation load. With the rough position provided by the INS, a local map is searched and selected from the pre-generated global map to improve the matching efficiency. NDT-based scan matching between the current Lidar frame and the local map is undertaken with the inertial-based initial Transformation Matrix. The Lidar-estimated vehicle pose can be obtained. A new real-time road map can also be generated if needed.

After obtaining the Lidar pose, the difference between the Lidar pose and the inertial propagation pose can be obtained, and the error within the inertial navigation information is estimated by the error-state EKF and then fed back to the inertial system to improve the pose results and bias estimation. When GNSS information is available, such as the RTK position results, these can also be used to correct the inertial navigation information to improve the accuracy and reliability of the localization system.

For data fusion of Lidar, INS, and GNSS, some current work has proposed using the graph optimization-based method to generate optimal localization and mapping solutions [179,257–259]. However, some of them are post-processed or highly dependent on GNSS data to mitigate the navigation drift, or even ignore the IMU bias. Since for our online Lidar/map matching based localization method, a reliable inertial navigation solution is essential to provide a good initialization for the scan/map matching process, and to increase the efficiency and accuracy of local map searching and selection, in-time IMU bias correction is critical and is more easily achieved using an EKF. As our test is undertaken within an urban area where GNSS signals are frequently lost, the feedback to the IMU states should also depend on the Lidar data, especially during GNSS outages. Moreover, the estimation uncertainty, which is an important parameter for the analyzing system solution, is seldom estimated in graph-based methods but can be directly estimated through the EKF method. Therefore in our current Method 2, the EKF method is used to fuse the Lidar/map localization, GNSS, and inertial navigation results. A comparison of Method 1 and Method 2 also highlights the difference between using an IMU sensor as a separate aiding sensor and as a tightly-coupled aiding sensor.

### 5.3.1. Estimation Results of Lidar/3D Map-Based Localization System

Since there is no ground truth information for this urban road test, the Lidar/map matching-based solutions are compared with the GNSS/INS solution within the three selected trajectory sections (Figure 14), during which the RTK status is "ambiguity-fixed" (Figure 6b).

Table 3 shows the comparison between the Lidar/map matching-based localization and the reference GNSS/INS localization results. For Method 1, the result differences fluctuate around zero, and their mean values are at the centimeter to decimeter level. The standard deviation for all three sections is around 0.1–0.2 m, therefore we treat the difference of the coordinates larger than 0.6 m as indicating possible outliers. The epochs that have outliers are about 1.7% of the total test data, which means the presence of outliers is rare. The possible reason for the outlier will be discussed in the next Section. For Method 2, the result has better accuracy. The standard deviation is around 0.05 m, much lower than that of Method 1. It can be seen from Figure 14, that Method 2 has a lower difference to the reference during the periods that Method 1 shows possible outliers, indicating that the tightly coupled method is more robust to outliers than simply using the INS solution for initialization.

### 5.3.2. Quality Analysis of the Numerical Results

The details of measurements during the epochs with big jumps (such as during the "red and green boxes" in Figure 14) are checked to investigate possible causes of the detected outliers. For Trajectory Section 1 in Figure 14 (red box), it is found that when driving around a roundabout, there were some big outliers by Method 1. The trajectory of the Lidar/map system and the GNSS/INS solution around this roundabout and their

views in Google Maps are shown in Figure 15. It can be seen that the GNSS/INS solutions (Figure 15 yellow line) are smoother in this area since the GNSS integer ambiguities are "fixed", while the Lidar/map solution has some differences to the reference trajectory if only using the IMU as a simple assistant (Method 1, Figure 15 blue line).

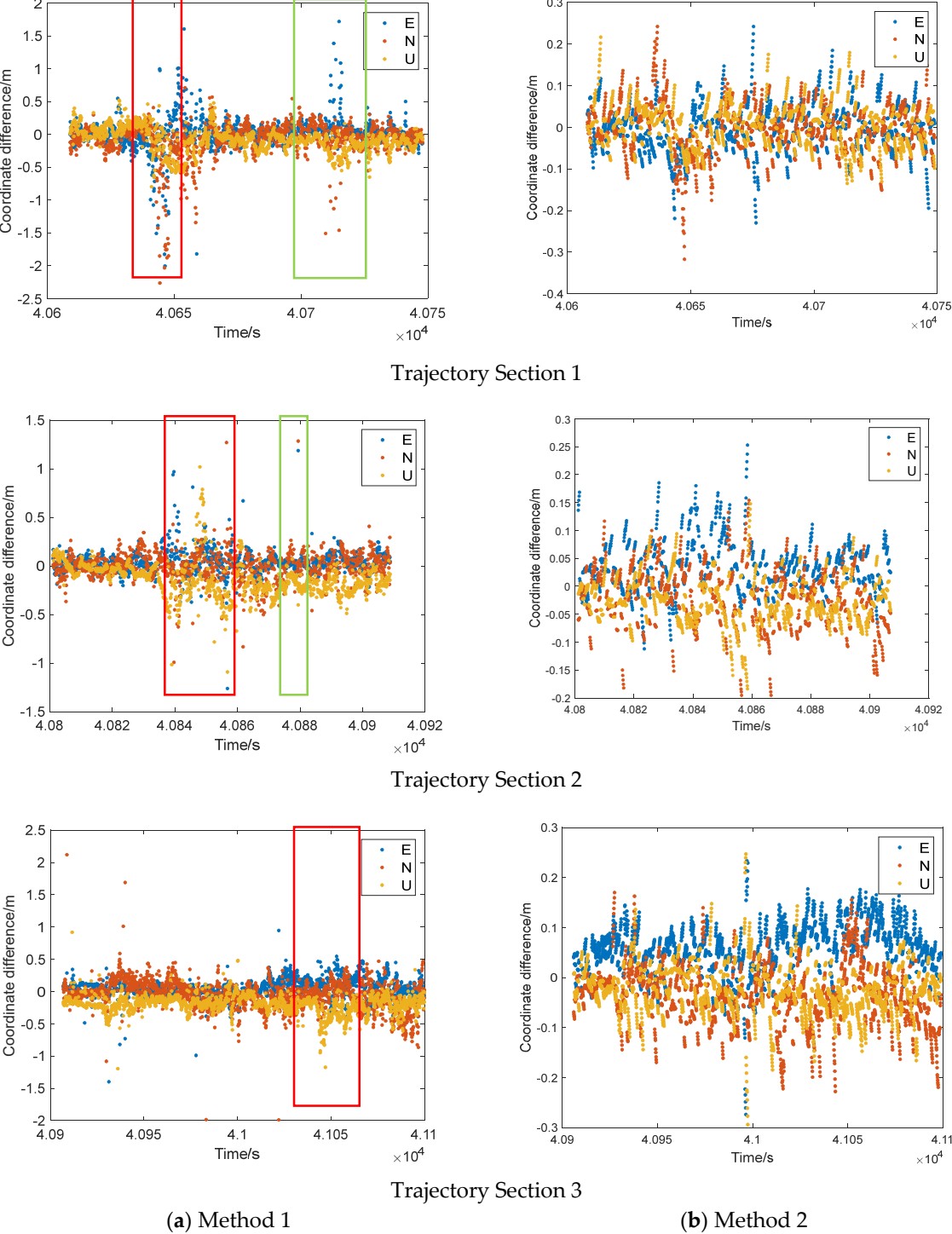

**(a)** Method 1      **(b)** Method 2

**Figure 14.** Coordinate difference between the proposed Lidar/map matching-based localization method and the reference GNSS/INS localization method at three trajectory sections. (**a**) Method 1: fusing IMU as an assistant sensor; (**b**) Method 2: fusing IMU using the EKF-based tightly coupled method. The red and green boxes indicate epochs with big coordinate difference with different timestamps.

**Table 3.** Mean and standard deviation for the difference between Lidar/map matching-based localization and the reference GNSS/INS localization results, for Trajectory Sections 1, 2 and 3 in Figure 14. (a) Method 1: fusing IMU as an assistant sensor; (b) Method 2: fusing IMUs using the EKF-based tightly coupled method.

| | | Trajectory Section 1 | Trajectory Section 2 | Trajectory Section 3 |
|---|---|---|---|---|
| | | Method 1 | | |
| | East | 0.020 | −0.036 | 0.051 |
| | North | −0.035 | 0.0031 | −0.048 |
| Mean (m) | Up | −0.084 | 0.140 | −0.189 |
| | | Method 2 | | |
| | East | −0.0026 | 0.0358 | 0.0571 |
| | North | −0.0052 | −0.0221 | −0.0371 |
| | Up | 0.0041 | −0.0250 | −0.0228 |
| | | Method 1 | | |
| | East | 0.142 | 0.099 | 0.128 |
| | North | 0.162 | 0137 | 0.188 |
| Stdev (m) | Up | 0.182 | 0.151 | 0.123 |
| | | Method 2 | | |
| | East | 0.0556 | 0.0466 | 0.0503 |
| | North | 0.0605 | 0.0530 | 0.0574 |
| | Up | 0.0481 | 0.0410 | 0.0486 |

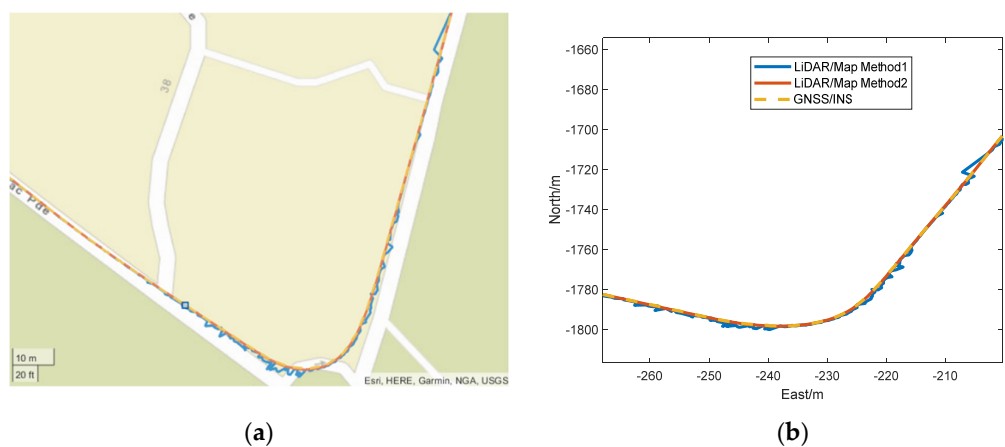

(**a**)      (**b**)

**Figure 15.** Trajectory of Lidar/map matching-based localization Method 1 (blue), Method 2 (Red) and GNSS/INS localization (Yellow): (**a**) view in Google Map; (**b**) view in local coordinate system around the roundabout.

Figure 16 shows the map view at this roundabout. It can be seen that the structure of the pre-generated map on the driving side of the road is not very clear since it lacks features around the trajectory. The roundabout is located at a parking area of a tourist attraction. There is no building and very few trees around this area. Since the testing was undertaken in the evening, there were not many parked vehicles that could be used as features. Therefore the quality of the matching step may be poorer, which results in degraded localization accuracy.

Figure 17 shows the Lidar scan view at this point with a range threshold of 20 m, and it can be seen that this Lidar scan does not have many usable features, especially after

pre-processing. Similar road environments with fewer features can be found when another cluster of outliers appeared in Trajectory Section 1 (green box), shown in Figure 14. In this situation, extending the range threshold may enhance the accuracy by including more features, however, it will increase the computational burden and be impacted by more outlier sources. Incorporating the inertial motion model by fusing the IMU more tightly may make the localization system more robust to this featureless condition (Figure 15 red line).

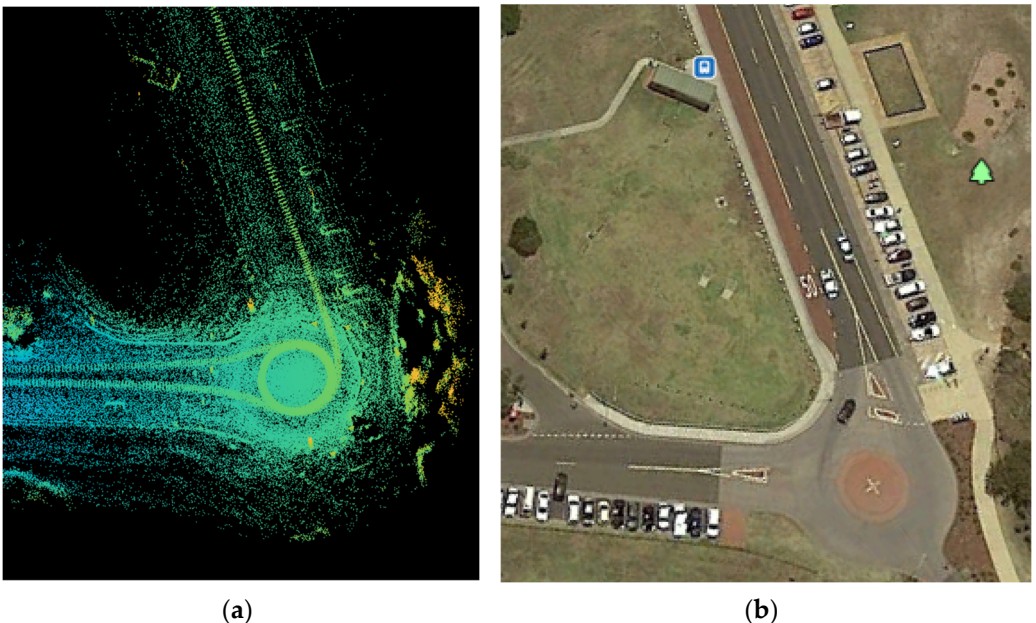

| (a) | (b) |

**Figure 16.** A section of (**a**) the generated map, and (**b**) the Google Earth view for the same location around the roundabout.

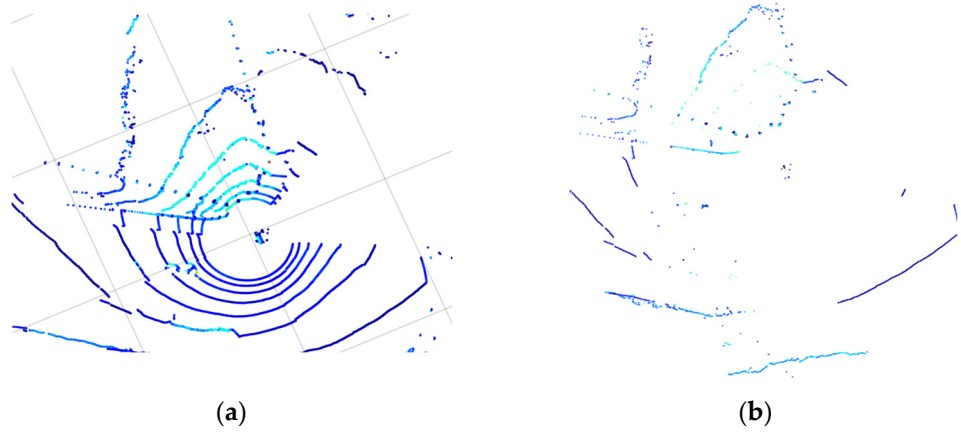

| (a) | (b) |

**Figure 17.** Scan frame at epoch 40,647 s with big outlier in Trajectory Section 1 around the roundabout ((**a**) the original scan view; (**b**) the view after pre-processing).

Another major source of outliers is the other moving entities around the host vehicle. Figures 18–20 show the Lidar views when there are big outliers in the localization stage.

No matter whether the other moving vehicle is on the same side of the road or the other side of the road, such a moving vehicle will influence the quality of the Lidar/map matching-based localization. When the moving vehicle is on the same side of the road as the host vehicle, it will result in bad estimation when it is initially detected or has a different speed from the host vehicle, or when it turns and drives onto another road. This will make this vehicle no longer detectable (red box in Figure 14 Trajectory Section 2 and Figure 18).

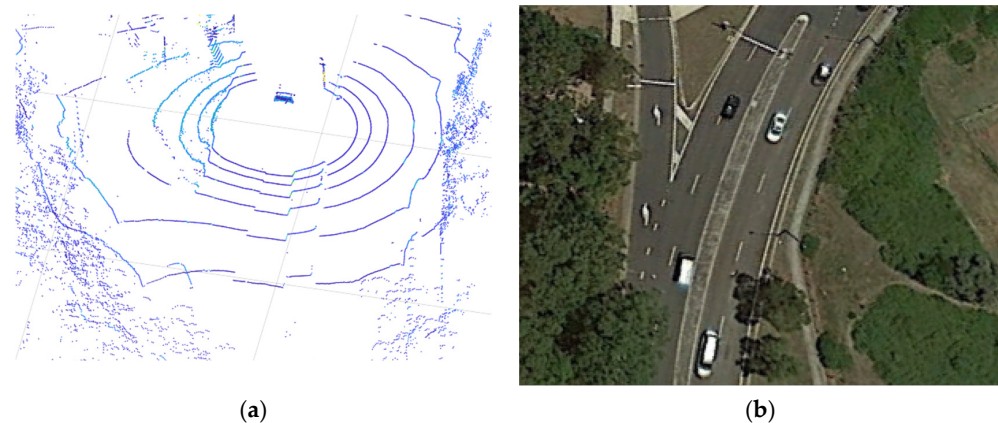

**Figure 18.** Scan frame at epoch 40,854 s with big outlier in Trajectory Section 2: a following vehicle is driving to another road, (**a**) Lidar scan; (**b**) Google Earth view.

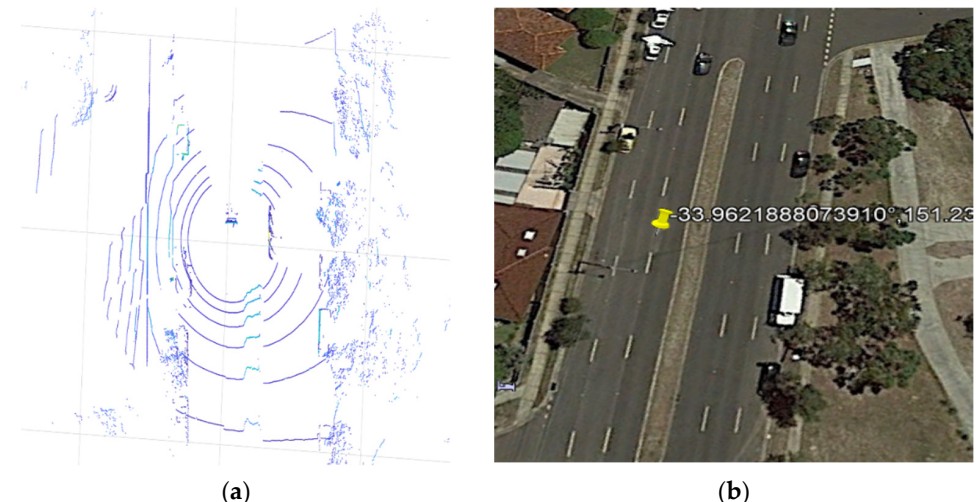

**Figure 19.** Scan frame at epoch 40,888 s with big outlier in Trajectory Section 2, an opposite driving vehicle detected, (**a**) Lidar scan; (**b**) Google Earth view.

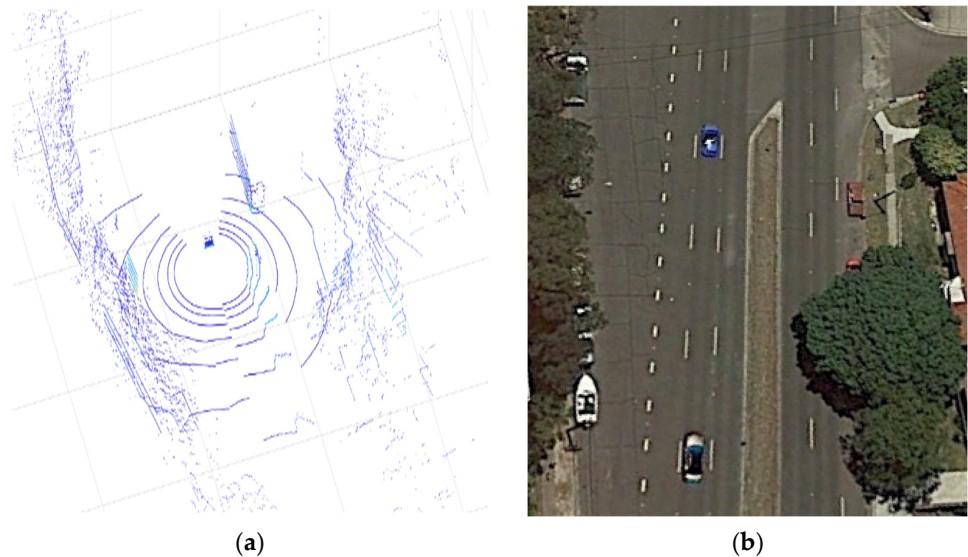

**Figure 20.** Scan frame at epoch 41,103 s with big outlier in Trajectory Section 3, with one tall bus driving past (**a**) Lidar scan; (**b**) Google Earth view.

Once the host vehicle detected an opposite driving vehicle, the localization estimation errors could reach 1–1.5 m (green box in Figure 14 Trajectory Section 2 and Figure 19).

The type of moving elements will also impact the presence of outliers. Most of the time, the vertical position estimation is less influenced by the moving elements. However, when checking the red box in Trajectory Section 3 (Figure 14), it can be found that the differences in the vertical direction are much higher than in other sections. By looking at the details of the Lidar view it is found that, at that section, a tall bus was driving past the host vehicle (Figure 20), which means vertical differences between the current Lidar scan and the pre-generated map might have caused some systematic vertical biases.

The moving objects within the road environment will be a major source of measurement outliers because the system treated the pre-generated map as a fixed reference map. Therefore, if there are any moving objects that cause the structures of the pre-generated map and the current scan frame to be different, outliers will occur. The moving objects, such as other vehicles, may exist in both the previous road mapping stage and in the current road scans for use in localization. For the step of offline HD map generation, such moving objects should be carefully identified and removed from the static 3D point cloud maps. For the online step, the moving objects could be identified and removed based on the cleaned pre-generated map, or directly achieved semantic segmentation with sensor data. Some researchers have developed some moving object segmentation methods, such as LMNet [166] which can distinguish moving and static objects based on CNN. Therefore, the possible detected moving objects could be removed, or the possible road environment change could be updated to the global map to enhance the accuracy of future driving around the same road path. These methods may be undertaken during the perception step. The aid of some numerical quality control methods may also contribute to this task at the localization and mapping steps, such as the FDI method, or outlier detection and identification methods, which can directly estimate and mitigate the influence of outliers from all kinds of resources, not only the moving outlier, and also other sensor or model faults.

Some FDI methods or integrity monitoring methods [255,256] have already been successfully applied to the GNSS/INS integration system under an EKF framework. Since in this case study the EKF method is used to fuse Lidar/map localization results and INS pose to generate high-frequency precise pose solutions, these quality control methods also indicate the potential for this proposed localization system. This will be a future research topic.

## 6. Conclusions

This paper gives a brief review of different SLAM approaches and their characteristics. SLAM has become a key approach for localization, mapping, planning, and controlling in automated driving. It shows promising progress in generating high-resolution maps for autonomous driving and for vehicle localization in road environments. The advantages and disadvantages of different SLAM techniques have been identified and their applications for autonomous driving have been discussed.

The trustworthiness of localization and navigation algorithms is an important issue for autonomous driving. There are many challenges that limit the performance of the SLAM techniques, which affect the safety of the localization and navigation results. These challenging issues, and possible solutions, are mentioned in this review. Furthermore, in order to ensure safety, the performance of the algorithms should be quantitatively evaluated with respect to such measures as accuracy, consistency, precision, reliability, and integrity. The methods to evaluate these qualities are briefly discussed.

A real-world road test was conducted to demonstrate the application of SLAM for autonomous driving with multi-sensor integration. The numerical results show that a GNSS/INS-aided-Lidar system can generate a georeferenced high-density point cloud map. This pre-generated map can then be used to support online localization, which has achieved about centimeter-level accuracy. This Lidar/map matching-based localization

method may also be useful to support an autonomous driving system during periods when GNSS signals are unavailable, which makes it suitable for urban area driving. A more tightly coupled fusion of IMU measurements will make the Lidar/map-based localization more accurate and robust to outliers than simply utilizing the inertial solution as assistant information.

Future studies should be focused on how to detect moving entities and mitigate their impact in the 3D point cloud mapping and localization process. In addition, integrity monitoring procedures for such Lidar/GNSS/INS-based vehicle localization and mapping system should be investigated.

**Author Contributions:** J.W. and S.Z. had the idea for the article. S.Z. performed the literature search, conducted the experiments and corresponding analysis, and wrote and revised the manuscript. J.W. reviewed and commented on the draft manuscript. W.D. supported the experiment setup. C.R. and A.E.-M. provided critical feedback. All authors have read and agreed to the published version of the manuscript.

**Funding:** This work is supported by the Australian Research Council (ARC) Project No. DP170103341.

**Data Availability Statement:** The data presented in this study are available on request from the corresponding author. The data are not publicly available due to the size of the data.

**Conflicts of Interest:** The authors declare no conflict of interest.

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
