# Peer review of "Simultaneous Localization and Mapping (SLAM) for Autonomous Driving: Concept and Analysis"

_remotesensing, doi:10.3390/rs15041156_

Round 1

Reviewer 1 Report (Previous Reviewer 1)

The authors did a lot of work to revise the paper. And they also add enough references. As for me, the current version is much improved compared to the last version. And I recommend it for publication now.

Author Response

Reviewer #1:

Comment: The authors did a lot of work to revise the paper. And they also add enough references. As for me, the current version is much improved compared to the last version. And I recommend it for publication now.

Response

Thank you for all the comments that have improved the quality of our paper.

Reviewer 2 Report (New Reviewer)

The paper under review aims to be botha review of the automotive SLAM and a description of of a localization method based on LIDAR-IMU-GNSS fusion with the use of a point cloud-based presurveyed map of the environment. The survey part is well written, and in principle is a comprehensive survey of the approaches to SLAM, concerning the algorithms/frameworks and the perception part. However, I found this part quite general, refering mostly to classic papers, while neglecting a large body of work on specific automotive SLAM/localization methods. In particular, the map representations conceived spcifically for automotive applications should be better exposed. Also the learning-based approaches that are recently in the focus are almost neglected. Please, update the paper by more recent references, perhaps giving up with some really old ones.

As to the new method proposed in the second part, it is described in the details, however, this part lacks a comparison to the state-of-the-art approaches, as fusion of LiDAR-based SLAM and GNSS is researched intensively in the automotive community. The proposed approach, which is based on the error-state Kalman filter seems to be rather standard with respect to the algorithm, and some advantages w.r.t. the factor graph algorithms that are used in the recent research need to be shown up in order to convince the reader about the paper's novelty.

Author Response

Attached please find the response letter.

Reviewer 3 Report (New Reviewer)

The manuscript is a review of different SLAM approaches, and a real experiment was carried out to demonstrate the application of SLAM for autonomous driving. The manuscript has been improved in this resubmission, and the reviewer thinks that all previous comments have been taken in account by authors.

The manuscript has been improved and now may be considered for publication in this journal 

Author Response

Reviewer #3:

Comment: The manuscript is a review of different SLAM approaches, and a real experiment was carried out to demonstrate the application of SLAM for autonomous driving. The manuscript has been improved in this resubmission, and the reviewer thinks that all previous comments have been taken in account by authors.

The manuscript has been improved and now may be considered for publication in this journal 

Response

Thank you for all the comments that have improved the quality of our paper.

Round 2

Reviewer 2 Report (New Reviewer)

As the my comments from the previous review have been addressed to a large extent, I recommend this paper for publication.

This manuscript is a resubmission of an earlier submission. The following is a list of the peer review reports and author responses from that submission.

Round 1

Reviewer 1 Report

There are lots of brilliant review articles about the SLAM technique even for the autonomous driving.

There are no valuable classification of the SLAM and some classification are ambiguous and controversial. For example, figure 1.1 , figure 2.1, and figure 2.2

 In addition, the related work is not current literature, some classification and conclusions are old and can be seen in many existing articles.

Specifically, the detained concerns are given below:

Page1 Line37, "an object" should correct to "a carrier"

Page2 Figure1.1 is not clear at all, positioning can also be treated as the perception process as the task of perception. The Camera/LiDAR are also can be used to estimate the pose of the carrier.

The difference and the connection between perception and positioning are key issues that this paper have to state.

Page3 Line110, "robust" should be "robustness'

Page5 Line178, there are lots of work address the inconsistency problem, please review them.

Page5 Line211, There are lots of  work about nonlinear optimization -based SLAM can also work online. 

Page6 Table1. the typical SLAM should be cited.

Page7 Line268, the fusion with imu is clear and the classification as loosely coupled and tightly coupled is not rigorous.

Page10 Line 402, there are lots of work reduce the linearization error of EKF SLAM , please review them and cite them.

Page10 Line 408, there are lots of work address the initial alignment problem.

Page11 Table2, possible solutions for the SLAM drifts are numerous, but there are few listed is the table. 

In fact, some problem is not obvious at all currently as some solutions are effective.

In short, there is no any innovation in the Section 5, meanwhile, as a review, these experiments are strange.

All the figures in section 5 should be rearranged.

Most of the trajectory of the experiment is under open skies, we believe that GNSS/INS is enough for the positioning.

Reviewer 2 Report

The paper reviews SLAM for autonomous driving including SLAM approaches, applications and challenges for autonomous driving, and a case study. The overall structure is well organized, and it is easy to follow. However, the paper quality would be improved if the following are revised appropriately.

(1) Localization is part of perception

In the first sentence of the second paragraph in Introduction, it says there are four key components for autonomous vehicles; localization, perception, planning and controlling. However, it is widely-accepted that autonomous driving consists of three components, perception, planning and control because localization is one of the sub-modules in perception. Therefore, the second paragraph is better to be revise accordingly.

(2) Variable case types

Lowercases and uppercases are mixed for variables in Figure 2.1 and 2.2. It may mislead readers that some are scalars and the others are vectors. Thus, it would be better to uniform the variable case types.

(3) Online vs Offline SLAM

Section 2.1 and 2.2 categorizes SLAM into online and offline, and matches filtering-based SLAM to online SLAM while optimization-based SLAM to offline SLAM. However, filtering can be used for offline SLAM and optimization can be used for online SLAM. Thus, it would be better to categorize into online/offline and filtering/optimization such as 2.1 Online SLAM, 2.2 Offline SLAM, 2.3 Filtering-based SLAM, and 2.4 Optimization-based SLAM.

(4) Inertial SLAM

Section 2.3 describes imu-integrated SLAM. However, multi-sensor fusion is a recent trend in SLAM. Thus, it would be better to revise it to Sensor Fusion for SLAM and include multiple sensors such as LiDAR, camera, IMU, and etc.

(5) Expressing the environment

The title of Section 4.2 would be better to be revised as "Representing" the environment. Also, only the point cloud representation is described. Other types of representations such as lines and planes should be considered.

(6) Tables and figures

Table 2 would be better to be revised wider. Figure 5.2 and 5.3 would be better to be combined in one figure. It would be better to check the accuracy if point clouds are overlaid on Google maps in the third sub-figures in Figure 5.9 and 5.13. The three sub-figures in Figure 5.11 would be better if they are placed in a row.

(7) Typos and better expressions

Line 274: efficiently -> efficient

Line 375: miniturized -> miniaturized

Line 441: vision SLAM -> visual SLAM

Line 469: Vehicle pose Root Mean Square Error (RMSE) -> Root Mean Square Errors (RMSE) of vehicle poses

Reviewer 3 Report

The article proposes a comparison of SLAM methods for the autonomous vehicle with the explanation of the concepts, an analysis of the current challenges and a real study of a LiDAR-based SLAM to build a map then a LiDAR-based loc on the built map.

SLAM has been a widely studied topic for more than thirty years with many methods and approaches using many different sensors (see article "A survey of state-of-the-art on visual SLAM"). The proposed state of the art is quite restricted, exposing only fusion methods (Kalman type or pose graph). But there is no mention of all the front-end methods able to use a whole set of sensors (camera, LiDAR, RADAR, GNSS, IMU...), mainly ICP based for LiDAR data like LOAM.

The difference presented between online SLAM and offline SLAM (Figures 2.1 and 2.2) is not correct. There are online SLAMs that will not only optimize the current pose X_{k+2} but also the previous poses. A SLAM graph (also called a pose graph) is not only offline. This can be used in the back-end of an online SLAM (in real time with a good implementation). The poses used (past poses or future poses) and the processing time (real time or not) that determines the online or offline method.

This article presents both SLAM methods (where the map is built online or offline) and Map-localization methods. This second task could deserve a survey in its own right (for example state of the art in "A Survey on Map-Based Localization Techniques for Autonomous Vehicles" with also dozens of different methods)

The authors offer an interesting discussion on the various current limitations of SLAMs and the challenges. But no discussion on the different metrics for accuracy, in particular the APE and RPE metrics (with or without alignment) which are an important point in the comparison of SLAMs.

We do not understand the objective of the experiment presented. The methods used (whether to build the map or to locate the map) are far from being state-of-the-art methods. The difficulties encountered have already been resolved in a number of methods.

In the experiments, a LiDAR Map-based localization method is tested. Finally, none of the SLAM methods presented (EKF fusion or particle filter) is really studied or compared.

Some othe remarks:

 - What is the accuracy of the GNSS/INS solution used as ground truth? -> map-based LIDAR+INS localization errors can be of the order of magnitude of GNSS+IMU based localization. 

 - No qualification of the point cloud map has been made

 - The method used as LiDAR localization is not well described and are not compared to state of the art

 - There seems to be no fusion method or at least taking into account the motion model of the vehicle which would probably avoid the outliers of Figure 5.12

 - No mention is made of the importance of deskewing lidar scans

 - There are now many MOD (Mobile Object Detection) methods that can avoid the various problems encountered during the actual study (Figure 5.15 or 5.16) (see article LMNet: Moving Object Segmentation in 3D LiDAR Data).